# Mechanistic insights into Bcs1-mediated mitochondrial membrane translocation of the folded Rieske protein

Cristian Rosales-Hernandez [ID], Matthias Thoms [ID], Otto Berninghausen [ID], Thomas Becker [ID] & Roland Beckmann [ID] [✉]

## Abstract

A functional mitochondrial respiratory chain requires coordinated and tightly regulated assembly of mitochondrial- and nuclear-encoded subunits. For bc1 complex (complex III) assembly, the iron–sulfur protein Rip1 must first be imported into the mitochondrial matrix to fold and acquire its 2Fe-2S cluster, then translocated and inserted into the inner mitochondrial membrane (IM). This translocation of folded Rip1 is accomplished by Bcs1, an unusual heptameric AAA ATPase that couples ATP hydrolysis to translocation. However, the molecular and mechanistic details of Bcs1-mediated Rip1 translocation have remained elusive. Here, we provide structural and biochemical evidence on how Bcs1 alternates between conformational states to translocate Rip1 across the IM. Using cryo-electron microscopy (cryo-EM), we identified substrate-bound pre-translocation and pre-release states, revealing how electrostatic interactions promote Rip1 binding to Bcs1. An ATP-induced conformational switch of the Bcs1 heptamer facilitates Rip1 translocation between two distinct aqueous vestibules—one exposed to the matrix, the other to the intermembrane space—in an airlock-like mechanism. This would minimize disruption of the IM permeability barrier, which could otherwise lead to proton leakage and compromised mitochondrial energy conversion.

**Keywords** Bcs1; Cryo-EM; Folded Protein Translocation; Mitochondria; Rieske
**Subject Categories** Membranes & Trafficking; Organelles; Structural Biology

## Introduction

In eukaryotes, 50–75% of proteins are transported to organelles or secreted outside the cell. To enter the endoplasmic reticulum (ER), mitochondria and chloroplasts, proteins need to translocate across or insert into lipid membranes. Here, transport is mediated by so-called translocators or translocons, which serve as protein-conducting channels. Typically, these translocons, such as the Sec complex in the ER or the Tim-Tom import machinery of mitochondria, recognize fully unfolded or only marginally folded (α-helical secondary structure) proteins as translocation competent substrates (for review see (Rapoport et al, 2017; Wiedemann and Pfanner, 2017)). Thereby, these translocons can accommodate a plethora of very different substrates. In contrast, the translocation of fully folded proteins is rather rare and is usually observed only as soon as folding and maturation to the native state is not possible in the final target compartment. For example, for many proteins that require insertion of complex cofactors such as iron-sulphur (Fe–S), iron-nickel (Fe–Ni) or molybdopterin clusters via enzymatic assembly factors, the maturation is completed before translocation (Berks, 1996; Santini et al, 1998; Sargent et al, 1998). In bacteria, chloroplasts and some *archaea*, such proteins are usually translocated by the twin-arginine translocase pathway (Tat pathway) (Frain et al, 2019; Palmer and Stansfeld, 2020). The main translocator unit is the small membrane protein TatA that has been suggested to oligomerize for cargo transport, but the exact mechanism still remains enigmatic. Mitochondria, however, lost the Tat system during evolution while still dealing with translocation of the highly conserved Fe–S cluster-containing Rieske protein (Rip1 in yeast). Rieske is part of the ubiquinol-cytochrome-c reductase, also known as bc1 respiratory chain complex or complex III in bacteria, mitochondria and chloroplasts (RIESKE et al, 1964) that is responsible for oxidation of the membrane pool of ubiquinol and reduction of cytochrome c in the mitochondrial intermembrane space (IMS) (Crofts, 2004; Xia et al, 2013). Rieske is a small integral membrane protein, with an N-terminal transmembrane helix (TMH) and a globular C-terminal domain bearing the 2Fe–2S cluster, which is exposed to the IMS (Harnisch et al, 1985). In mitochondria, Rieske has a dedicated translocator named Bcs1 (bc1 synthesis 1) (Aldridge et al, 2008; Bachmann et al, 2006; De Buck et al, 2007) which is a member of the AAA-ATPase family. While members of this class usually are enzymes that couple ATP hydrolysis with distinct unfolding or disaggregation activities, Bcs1 belongs to an outlying clade (Frickey and Lupas, 2004) and functions as a translocator for folded proteins.

Mitochondrial assembly of yeast Rip1 and its insertion into the bc1 complex is a multistep process (Wagener and Neupert, 2012). The Rip1 precursor carries a mitochondrial matrix targeting signal and is

Department of Biochemistry, Gene Center, University of Munich, Feodor-Lynen-Str. 25, 81377 Munich, Germany. ✉E-mail: beckmann@genzentrum.lmu.de

initially imported into the matrix of mitochondria via the general import pore by TOM and TIM23 complexes (Hartl et al, 1986; van Loon et al, 1987). Subsequently, the 30 amino acid long mitochondrial targeting pre-sequence is removed in the matrix in two processing steps (Graham and Trumpower, 1991) and the C-terminal domain folds and acquires its 2Fe–2S cluster (FeS domain) (Kispal et al, 1999; Wagener et al, 2011). During late stages of assembly, Rip1 can associate with the small protein Mzm1 (Cui et al, 2012), which was shown to prevent aggregation and proteolytic decay of Rip1 in the matrix. Finally, the completely folded Rieske C-terminal domain translocates from the matrix to the IMS while the N-terminal TMH inserts into the inner mitochondrial membrane (IM) (Wagener et al, 2011), leading to bc1 complex maturation. Initially, it was suggested that incorporation of Rieske protein into the bc1 complex by Bcs1 could be a prerequisite for the dimerization of bc1 and super complex formation (Cruciat et al, 2000). More recently, it has been shown that dimerization and further super complex formation can be observed in the absence of Rip1 (Conte et al, 2015; Stephan and Ott, 2020).

First structural insights into the architecture of the Bcs1 complex were gained from cryo-EM and X-ray crystallography studies of yeast (Saccharomyces cerevisiae) and mouse (Mus musculus) Bcs1 (yBcs1 and mBCS1, respectively) (Kater et al, 2020; Tang et al, 2020). Here, contrary to most AAA-ATPases that form hexameric rings with rather narrow central pores, Bcs1 was shown to form heptameric rings with a significantly wider pore and two large vestibules. This characteristic heptameric architecture is a result of specific features of the Bcs1 protomers. One protomer consists of three domains, a transmembrane helix (TMH) at the N-terminus, a Bcs1-specific β-sheet-containing middle domain and a C-terminal AAA cassette (Kater et al, 2020; Tang et al, 2020). Upon assembly, the seven TMHs form the transmembrane domain (TMD), a basket-like structure in the IM establishing a large and partly aqueous space in the hydrophobic lipid bilayer (also called IM vestibule). Towards the matrix side, the middle domains are assembled into a proteinaceous ring that forms a seal-like structure (seal pore), whereas the seven AAA cassettes form a second cavity opening to the matrix side (matrix vestibule).

Structural analyses have shown that the arrangement of the individual domains with respect to each other can undergo changes leading to global rearrangements of the Bcs1 quaternary structure, giving rise to the idea of an airlock-like translocation mechanism for Rieske. The pre-translocation state, as represented by structures of the ADP-bound state of yBcs1 (Kater et al, 2020) as well as ADP- and Apo states of mBCS1 (Tang et al, 2020) showed a large matrix vestibule formed by AAA and middle domains that would be wide enough to accommodate the folded FeS domain of Rip1. These structures also show the seal between the matrix and the IM vestibules to be closed. The two yeast Apo states both show a compaction of the matrix vestibule but differ in the conformation of the middle domain that leads to an opening of the seal pore. When bound to ATPγS as observed with mBCS1 (6UKS and EMD-20811) the size of the matrix vestibule is further reduced while the seal pore is opened, indicating that ATP binding may trigger transient opening to gate Rieske from the matrix to the IM vestibule. Notably, in both yBcs1-Apo2 and mBCS1-ATPγS structures, the basket-TMHs are largely disordered, indicating that the basket may open both vertically to release the FeS domain and laterally to release the Rieske TM for integration into the adjacent bc1 complex.

This proposed mechanism was further corroborated by a high-speed atomic force microscopy (HS-AFM) and line scanning (HS-AFM-LS) study (Pan et al, 2023) using purified mBCS1 in different nucleotide states (ATPγS, ADP, apo), confirming the nucleotide-dependent conformational changes and manifesting the concerted nature of such changes in Bcs1. Yet, for a long time, how substrate recognition, translocation and membrane insertion are coupled to the Bcs1's ATPase cycle remained enigmatic due to a lack of substrate-bound Bcs1 structures. Only recently, a low-resolution cryo-EM structure of mBCS1 bound to bovine Rieske FeS domain (Iron-Sulphur Protein extrinsic domain; ISP-ED) gave first insights into the substrate recognition step (Zhan et al, 2024). Here, a bulky extra density was indeed found inside the matrix vestibule, asymmetrically bound to one or two protomers of the Bcs1 heptamer that adopted the ADP-bound conformation of mBCS1. Yet, limited resolution did not allow to determine the orientation of the ISP-ED of Rieske or to unambiguously assign the nucleotide state in Bcs1.

Thus, several key questions regarding the Bcs1-mediated Rieske translocation mechanism remain open. First, it is unclear in which conformational state of Bcs1 substrate recognition takes place. Second, the molecular basis of substrate recognition and accommodation remains to be established. This is especially important with respect to positioning of the Rieske TM that needs to be inserted into the IM in an $N_{Matrix}$-$C_{IMS}$ topology. Third, so far, no structural information on the actual gating step from the matrix vestibule into the IM vestibule is available which would be required to allow the transition of the Rieske protein from the matrix into the IM vestibule.

Here, we present cryo-EM structures of in vitro reconstituted yeast Bcs1-Rip1 complexes under various conditions. We observed in a 3.4 Å resolution structure that Rip1 binds to Bcs1 in the Apo state. We could position the Rip1-FeS domain as well as parts of the TM segment inside the Bcs1 matrix vestibule based on resolved secondary structure. This further allowed us to map residues in Bcs1 and Rip1 crucial for substrate accommodation, mutation of which leads to respiratory growth defects. We further show structures of Bcs1 in two distinct ATPγS states, which display a largely constricted matrix vestibule. We could visualize Rip1 in one of the two states, relocated through the wide-open seal pore into the IM vestibule. We conclude that ATP binding to substrate-loaded Bcs1 leads to compaction of the matrix vestibule and, at the same time, to gating of the substrate into the IM vestibule, accompanied with a conformational change of the middle domains that results in opening of the TMD helices.

# Results

## Structure of Bcs1-Rip1-FeS in the substrate loading state

To obtain stable Bcs1-bound Rip1 translocation intermediates, we chose an in vitro reconstitution approach using purified components. Therefore, we first developed a protocol for rapid affinity purification of Bcs1 solubilized in digitonin from S. cerevisiae (Appendix Fig. S1A,D). For Rip1, we chose two constructs (Appendix Fig. S1B,C), one containing only the C-terminal FeS domain but lacking the N-terminal TM segment (Rip1-FeS; aa 83–215) and one that still contains parts of the TM (Rip1-TM; aa

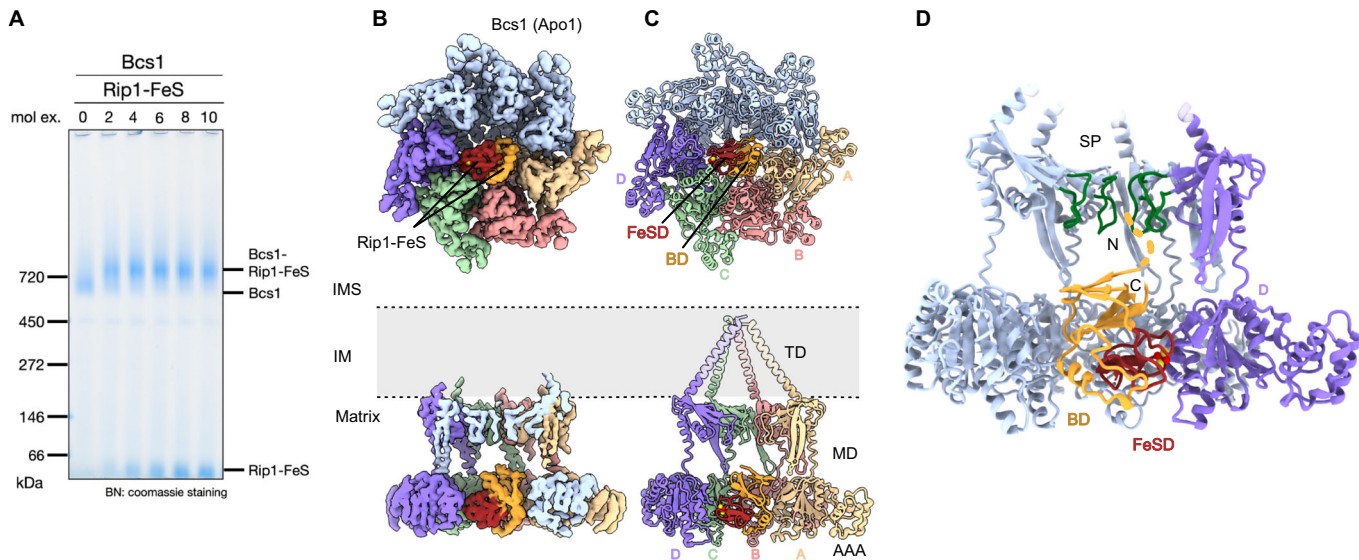

**Figure 1. Structure of a Bcs1-Rip1 substrate loading complex.**

(A) Blue native (BN) gel showing the in vitro reconstitution of the Bcs1-Rip1 complex. Purified Rip1-FeS was added to purified Bcs1 in the absence of any nucleotide in increasing molar excess as indicated. (B, C) Cryo-EM structure (B) and molecular model (C) of the Bcs1-Rip1-TM complex. Upper panels are bottom views from the matrix side, lower panels cut side views to highlight the Rip1 position. The transmembrane basket helices were flexible in our maps, and the model shown represents their position in the Apo1 conformation (Kater et al, 2020). They are shown transparent to indicate that they are not present in our map. Protomers interacting with Rip1 are color-coded and labeled from (A–D). (D) Orientation of Rip1-FeS in the Bcs1 matrix vestibule. The N- and C-terminus of Rip1 are facing the seal pore (SP) and the dashed line indicates the putative position of N-terminal residues (including the TM helix). IMS intermembrane space, IM inner membrane, TD transmembrane domain, MD middle domain, AAA CAA-ATPase domain, BD base domain, FeSD 2Fe–2S cluster binding subdomain. The 2Fe–2S cluster is shown in a sphere representation, where yellow spheres represent the sulfur atoms and red spheres represent the iron atoms. Source data are available online for this figure.

31–55 and 66–215) and that was shown to be successfully integrated into the bc1 complex (Ramabadran et al, 1997). Both proteins were expressed in *E. coli* and affinity purified using a twin Strep tag (Appendix Fig. S1E,F). Bcs1 heptamer and Rip1 were further purified by size-exclusion chromatography and Bcs1 was checked for quality and oligomerization state by negative stain TEM (Appendix Fig. S1D). Binding of Rip1 to Bcs1 was monitored by a size shift of the apparent molecular weight in Blue native gel electrophoresis (BNGE) as described before (Wagener et al, 2011) (Fig. 1A; Appendix Fig. S1E,F). This shift (from about 700 to 800 kDa) occurred as Rip1-FeS or Rip1-TM were added in at least twofold molar excess over Bcs1 in the absence of any nucleotide.

Accordingly, cryo-EM samples were prepared from in vitro reconstituted Bcs1-(Rip1-FeS) and Bcs1-(Rip1-TM) complexes and subjected to Single Particle Analysis. 3D reconstruction of both samples yielded classes with clear Rip1 density within the Bcs1 matrix vestibule (Appendix Figs. S2 and S3). Those classes (of 106,497 particles for Bcs1-(Rip1-FeS); of 57,257 particles for Bcs1-(Rip1-TM)) were refined to a final resolution of 3.5 Å and 3.4 Å, respectively with local resolution ranging from 3.0 to 6.5 Å (Appendix Fig. S4). This allowed us to unambiguously identify the Rip1-FeS domain (30 Å in diameter) and determine its orientation within the Bcs1 matrix vestibule, resulting in a near-complete molecular model for a stable Bcs1-Rip1 pre-translocation intermediate (Fig. 1C).

Both Bcs1-(Rip1-FeS) and Bcs1-(Rip1-TM) complexes displayed a very similar overall architecture, with Bcs1 adopting the same conformation as observed in Apo1 state of yBcs1 (Kater et al, 2020) (Figs. 1B and EV1). This state adopts a closed seal pore, but the

matrix vestibule was already somewhat more compacted when compared to the ADP state. In addition, no ADP density was found in the ATP-binding pocket of any of the seven protomers (Fig. EV2A,B). Notably, the substrate loading state clearly differs from mBCS1, where Apo and ADP states adopt the same conformation (Tang et al, 2020; Zhan et al, 2024). Since the Bcs1-(Rip1-TM) reconstruction showed a slightly better resolution and overall map quality, it was used for molecular model building and the detailed structure will be discussed based on this map (Appendix Table S1).

Rip1-FeS introduces asymmetry in the matrix vestibule by exclusively contacting the AAA domains of four Bcs1 protomers, henceforth called protomers A–D (Fig. 1C). It binds to Bcs1 with the 2Fe–2S cluster binding subdomain (FeSD) tightly packing against protomers C and D while the large (base) domain (BD) is exposed to protomers A and B (Fig. 1C). Based on resolved secondary structure including several side chains in the Rip1-FeS density (Appendix Fig. S4A, left panel and Fig. EV3), we could position the FeS domain with its N- and C-termini facing towards the seal pore locking up the IM vestibule. In this orientation, the N-terminal TM helix would also face towards the IM vestibule (Fig. 1D).

In Bcs1, we identified two highly conserved charged motifs, the EWR motif (E212-R214) and the DDR motif (D300-R302), that contain residues exposed to the FeS domain of Rip1 (Fig. 2A, left panel; Appendix Fig. S5). The EWR motif was located in the Bcs1-specific β-strands β-a1 and β-a2 (BSB) that extend the canonical six β sheets in the AAA domain, whereas the DDR motif was located at the tip of helix α2, following one of the loops (pore loop 1 (PL1)) corresponding

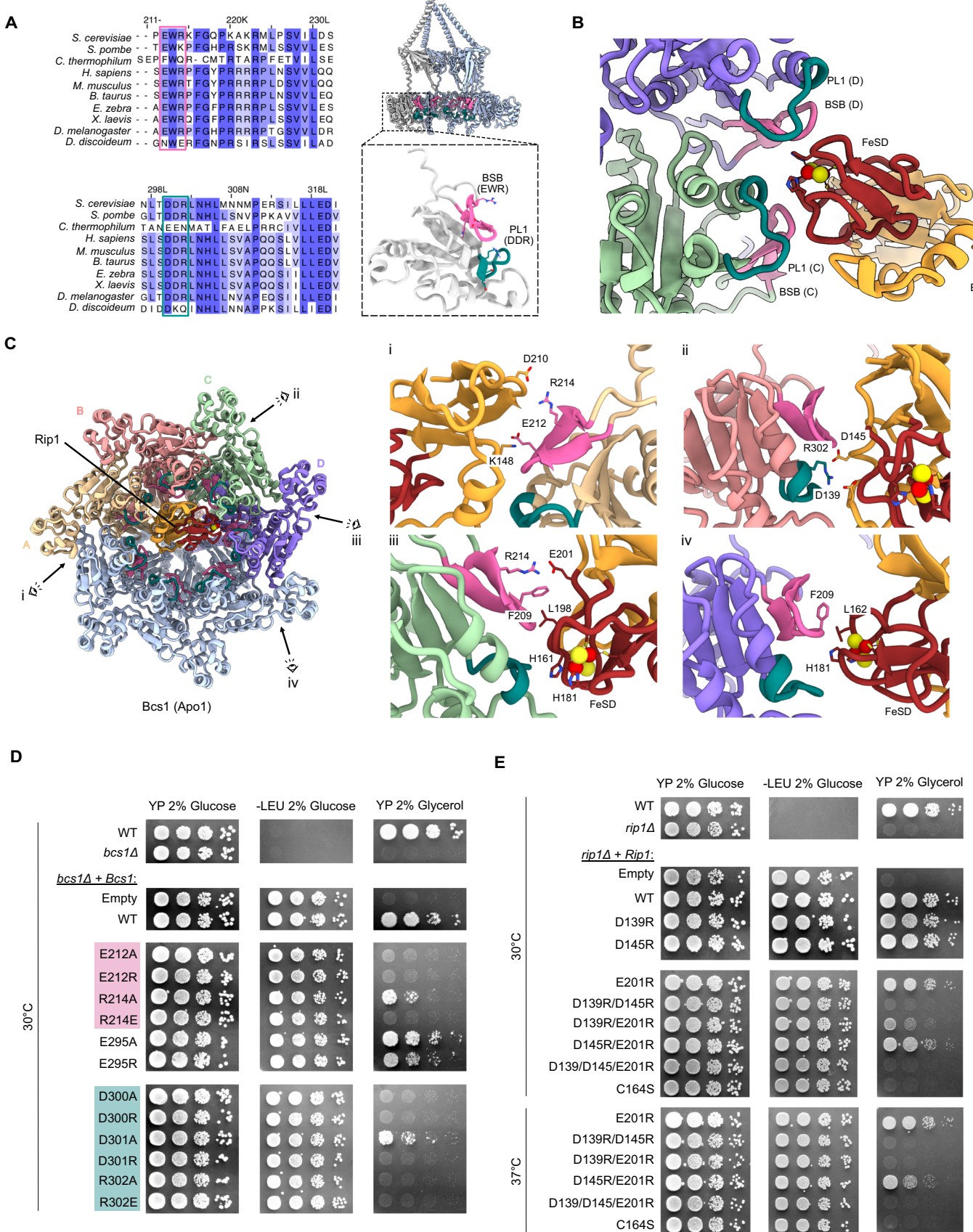

**Figure 2. Conserved motifs mediate the interaction between Bcs1 and Rip1-FeS.**

(A) Multiple sequence alignment displaying two conserved motifs (EWR motif and DDR motif) and their location in the Bcs1 protomer. (B) Close-up view highlighting the interaction between Bcs1 protomers C and D with the Rip1-FeSD. (C) Left panel: Bottom view on the Bcs1-Rip1-TM complex. View directions for close-up views (i-iv; right panels) are indicated. Right panel: Zoom on the interactions of protomers A (i), B (ii), C (iii), and D (iv) with Rip1-FeS. Side chains of interacting residues, including the conserved EWR motif (E212 and R214) and the DDR motif (R302), are indicated. The 2Fe–2S cluster is shown in a sphere representation, where yellow spheres represent the sulfur atoms and red spheres represent the iron atoms. (D) Mutational analysis of Bcs1 residues in EWR and DDR motifs based on a growth assay (tenfold serial dilutions) of yeast cells on a fermentable (glucose) or non-fermentable (glycerol) carbon source. Right panel shows the growth control on rich media (YP), middle panel is a control for the presence of the Bcs1-expressing plasmid, left panel shows growth on YP with glycerol. Growth was monitored at 30 °C unless indicated. (E) Growth assays for Rip1 were carried out in a similar way on a $rip1\Delta$ background and from a Rip1-expressing plasmid (see "Methods" for details). (D, E) Are an excerpt of the comprehensive combined display shown in Appendix Fig. S6, reduced here to emphasize the residues discussed in the text and shown in (A–C). Source data are available online for this figure.

to the substrate-binding pore loops in canonical AAA-ATPases (Fig. 2A, right panel). First, we observed that the BSB and PL1 from protomers C and D are forming contacts to the Rip1-FeSD (Fig. 2B). Second, the EWR and DDR motifs—together with F209—formed a module that contributes to substrate contacts in all four interacting protomers (Fig. 2C). Thereby, a characteristic positive charge distribution pattern is established in the matrix vestibule that is complementary to several negatively charged residues, e.g., D139, D145, E201 and D210 in the Rieske substrate. However, in each protomer, these residues can exhibit a different binding mode complementary to the surface properties of the Rip1-FeS.

We next tested whether the above-described charged residues in Bcs1 and Rip1 are indeed involved in transient complex formation and are thus important for Bcs1-mediated translocation. We expressed Bcs1 and Rip1 mutants under the control of Bcs1 endogenous promotor in a $bcs1\Delta$ and a $rip1\Delta$ yeast strain while checking for respiratory growth defects in a non-fermentable carbon source. Indeed, we found that Bcs1 charge inversion mutations in the EWR and the DDR motives were not viable in YP medium containing glycerol. For E212, D300 and R302, also mutations to alanine were lethal in respiratory conditions, whereas R214A and D301A show only a mild effect. In contrast, mutants of an unrelated residue in the vicinity of the DDR motif show no lethal effect (E295, located in PL1) (Fig. 2D; Appendix Fig. S6A). For Rip1, we find that single charge inversion mutations (D139R, D145R, E201R and D210R) did not show any growth defect while double mutations indeed show lethal phenotypes on a non-fermentable carbon source (Fig. 2E; Appendix Fig. S6B). To support our hypothesis that a charge complementarity between Rip1 and Bcs1 is required for Rip1 translocation, we performed a multiple sequence alignment of Rip1 homologs in species that use both Bcs1 (in mitochondria) and the Tat pathway (petC in chloroplasts or qcrA in prokaryotes). The alignment shows that D139, D145, E201 are conserved only in cases where Bcs1 is required for translocation (Fig. EV4A,B). In contrast, only D139 is conserved in chloroplasts, and none of the three residues are conserved in prokaryotes. Moreover, we note that these residues are exposed to the surface when Rip1 is integrated into complex III, indicating that they are not required for complex III activity (Fig. EV4C–E).

## Bcs1 undergoes conformational changes upon nucleotide binding

Our observation of the pre-translocation state in the absence of any bound nucleotide (Fig. EV2A,B) suggests that subsequent binding

of ATP or ATP hydrolysis may trigger substrate translocation. This question was addressed using BNGE-based binding assays as shown above. When incubated in presence of nucleotides, Bcs1 exhibits characteristic migration shifts, which can be correlated with conformational changes that lead to different electrophoretic mobilities in the gel. After incubation with ADP or ATP (for 10 min), we observed a similar size shift (from about 700 to 750 KDa) in BNGE as upon addition of Rip1-FeS (Figs. 1A and 3A). Here, we assume that, in the case of ATP, the resulting band represents also the ADP state, since Bcs1 hydrolyzes ATP in vitro at an average rate of 27 nmol ATP / nmol yBcs1 (heptamer) / min (Appendix Fig. S1G). The observed size shift thus reflects the conformational change of Bcs1 from the Apo state to the ADP state. Interestingly, ATPγS incubation caused a smaller but still clearly discernable size shift when compared to ATP or Rip1-FeS, indicative of another distinct conformation of Bcs1.

To test this hypothesis, we determined the structure of Bcs1 bound to ATPγS by cryo-EM. Here, we were able to distinguish two conformations of Bcs1 after 3D classification that differed from previously observed yBcs1 classes. These two subsets (here termed ATPγS1 and ATPγS2) were refined to an overall resolution of 2.6 and 2.7 Å, respectively (local resolution ranging from 2.2 to 5 Å for both ATPγS states (Appendix Figs. S7–S9). Both reconstructions showed Bcs1 in an entirely different, more compacted conformation when compared to the other observed nucleotide states (Fig. 3B) from yeast (Bcs1-Apo1 and Bcs1-ADP), but its overall architecture was similar to mBCS-ATPγS (Tang et al, 2020). ATPγS density was observed in each of the seven protomers as well (Fig. EV2C,D). Notably, our reconstructions showed a substantially narrowed matrix vestibule of ~20 Å in diameter as opposed to ~40 Å and ~30 Å in the ADP or apo states, respectively. This dimension is incompatible with substrate binding at the matrix side and coincided with a wide-open seal ring of 30 Å diameter (only 15 Å in the ADP and Apo1 states) large enough to accommodate Rip1 during translocation (Fig. 3B). Moreover, the basket TM helices are in a defined position and clearly resolved in ATPγS1, occupying a similar position as observed in yeast ADP or Apo1 states, tightly held together by a hydrophobic seal in IMS leaflet of the IM. In ATPγS2, however, the TM domain becomes more flexible, especially in the hydrophobic seal region (Fig. 3B; Appendix Fig. S8A).

For both of our ATPγS states, the arrangement of the AAA-ATPase domains relative to each other was significantly different from yBcs1-Apo1 and yBcs1-ADP (Fig. 3C). The observed inward rotation for the transition between ADP- and Apo state is even more pronounced for both ATPγS states (of approx. 25° from

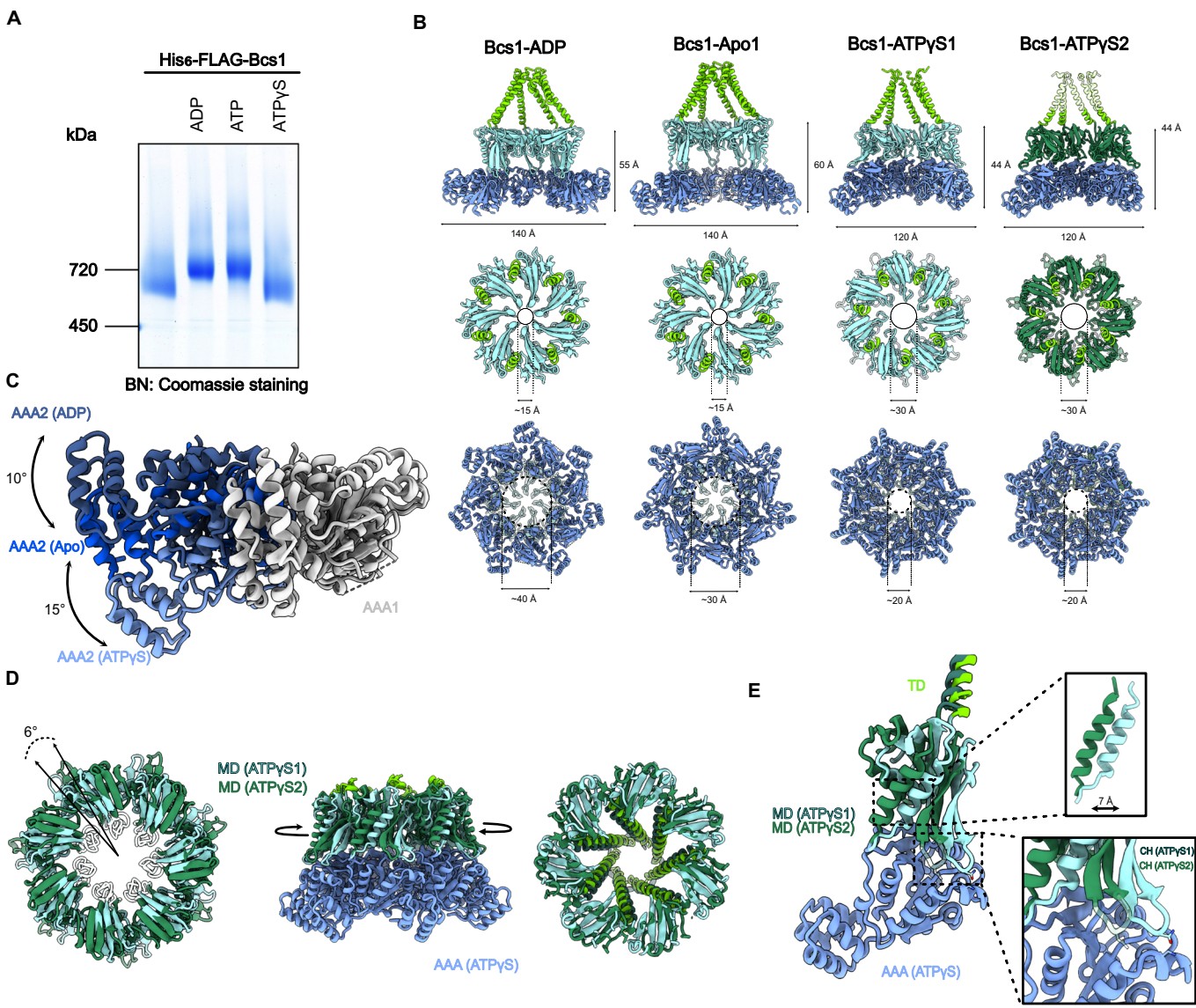

**Figure 3. Conformational dynamics of Bcs1.**

(A) Blue native (BN) gel showing the Bcs1 oligomer after incubation with ADP, ATP and ATPγS. (B) Comparison of molecular models for Bcs1-ADP, Bcs1-Apo1 (Kater et al, 2020) as well as two states of Bcs1 bound to ATPγS (ATPγS1 and ATPγS2). Upper panels show cut side views, middle panels show top views from the IMS side at the middle domains and lower panels bottom views from the matrix side at the AAA domains. Note the compaction of Bcs1, the opening of the seal pore and the narrowing of the matrix vestibule in the ATPγS states. (C) Conformational rearrangements of the AAA domain from the ADP via Apo1 to ATPγS states: when aligned to the AAA domain of one protomer (AAA1) the neighboring AAA domain undergoes in inward rotation (10° from ADP to Apo1; 15° from Apo1 to ATPγS) relative to AAA1. (D) Overlay of the two ATPγS states: The ring formed by the middle domains rotates versus the AAA ring by 6° as shown in a top view (left panel) and a side view (right panel). The seal loops are shown transparent, indicating that they are largely delocalized. (E) Comparison of the two ATPγS states in one isolated protomer. The middle domain shifts with respect to the AAA domain by 7 Å. In ATPγS1 the CH of the neighboring protomer forms a contact with the large AAA domain involving E124, N125 (CH) and K250, S253 (AAA). In ATPγS2 the CH dissociates, and the loop becomes more flexible as indicated by transparency. Source data are available online for this figure.

ADP- to -ATPγS- and 15 degrees from Apo1- to ATPγS state). For the middle domains, we observe a similar tilt motion away from the membrane plane as observed for the transition from ADP (or Apo1) to the Apo2 state (Appendix Fig. S10). We also observed the outward movement of the connector hairpin (CH) that can adopt two positions (see below). Thus, as also observed in the Apo2 state, both ATPγS states show a wide opening of the seal pore (res

161–168) with seal loop residues being largely delocalized in these structures.

The two ATPγS states can be distinguished from each other mainly by the orientation of the middle versus the AAA domains (Fig. 3D). While the conformation of the AAA domains is essentially the same, we observed a rotation of the middle domains of 6° degrees against the AAA domains, leading to a shift of the

middle domains by 7 Å away from the AAA domain in each protomer (Fig. 3E). Notably, the same shift and rotation also apply to the TD helices that rotate with the middle domains as rigid bodies (Fig. 3D, right panel). This rotation may also lead to destabilization of the TD, which would explain why its helices are less defined in the ATPγS2 state.

Of note, in ATPγS1, the CH forms a contact with the large AAA domain of the neighboring protomer (D124/N125 of the CH with K250 and S253 of AAA), but this contact breaks after the transition to ATPγS2, concomitantly leading to a higher flexibility of the CH loop (Fig. 3E; Appendix Fig. S9A). A direct contact between the CH and AAA domain was also observed for the ADP state, here involving N287 and R313 of the AAA (Kater et al, 2020). This suggests that the conformation of the CH may be an important signal for the status of translocation and that the CH may transmit changes in the MDs, e.g., due to the presence of substrate or unpacking of seal loops, to the AAA domains. In our structures, however, in the ATPγS-bound nucleotide binding site we do not detect any significant differences when comparing the AAA domains in both states.

Taken together, yBcs1 in the ATPγS state shows a severely contracted matrix vestibule — as observed for mBCS1 — which is too small to bind the substrate in the loading state. However, this closing of the matrix vestibule together with the widening of the seal pore suggests that the two observed ATPγS conformations may also occur at intermediate stages of substrate gating, as also proposed before for mBCS1 (Pan et al, 2023; Tang et al, 2020; Zhan et al, 2024).

## ATPγS locks the Bcs1-Rip1 complex in a substrate gating intermediate

To establish conditions for the stabilization of the Bcs1-Rip1 translocation intermediate, we performed binding assays as described above using Rip1-FeS and Rip1-TM with ATP and non-hydrolysable ATPγS and varied the order of component addition for complex assembly (Fig. 4A). When adding Rip1-FeS followed by ATP, the BNGE bands became rather smeary indicating multiple conformations, probably due to a mixture of states occurring at various stages of translocation (Fig. 4A, lane 5). We observed a similar smeary appearance of the bands when first adding ATP for 10 min and then Rip1-FeS (Fig. 4A, lane 7). When adding ATPγS followed by Rip1-FeS (Fig. 4A, lane 8), no further size shift was observed when compared to adding only ATPγS (Fig. 4A, lane 4), indicating no or only weak binding of Rip1-FeS to Bcs1 in the ATPγS state, which is in line with our observation that the matrix vestibule is constricted in this state and cannot accommodate the substrate. However, we saw a clear difference when Rip1-FeS was added first, followed by ATPγS (Fig. 4A, lane 6). Here, the bands became again rather smeary with no clear shift anymore towards higher apparent molecular weight. We conclude that after Rip1-FeS binds to Bcs1 the addition of ATPγS induces a conformation of Rip1-FeS-bound Bcs1 that is different from the one observed for Bcs1-Rip1-FeS in Apo state and different from the Bcs1-ATPγS. We hypothesized that Rip1 binding to Bcs1 is required prior to ATP, e.g., in the Apo state as shown above or in the ADP state. Addition of a non-hydrolysable ATP analog before incubation with substrate most likely locks Bcs1 in a constricted conformation entirely preventing access of Rip1 to the Bcs1 matrix vestibule.

We thus structurally analyzed the pre-formed Bcs1-Rip1 complex incubated with ATPγS by cryo-EM. Single particle analysis showed Bcs1 again in the two ATPγS states and, in addition, we obtained one class (70,366 particles) that displayed extra density for Rip1-FeS in the IM vestibule (Fig. 4B), refined to an overall resolution of 3.1 Å (local resolution ranging from 2.7–7.5 Å) (Fig. 4B; Appendix Figs. S8 and S9). Interestingly, we exclusively observed Rip1 bound to Bcs1 in the ATPγS2 conformation, with all seven ATP-binding pockets exhibiting ATPγS density (Fig. EV2E), the only difference to the unbound ATPγS2 state being that the TD basket helices were now completely delocalized. We find an extra density of ~30 Å in diameter for Rip1 in the aqueous cavity between the middle domain and the TD basket and it appears to be, at least partly, clogging the seal pore from the IM side, indicating that the substrate itself could play a role for the maintenance of the permeability barrier during and immediately after translocation. Notably, however, in contrast to the loading state, the density for Rip1-FeS in the IM vestibule did not display any obvious orientation preference, as confirmed by a rather low local resolution. Moreover, we observed that the central loops of the middle domain appear more delocalized compared with the unbound ATPγS2 state. We thus were not able to position the Rip1-FeS domain in a distinct orientation, and no specific contacts with Bcs1 could be identified.

We conclude that binding of ATPγS to a pre-formed Bcs1-Rip1 complex triggered a conformational change in Bcs1, i.e., matrix vestibule constriction and opening of the seal pore, that results in translocation of the substrate from the matrix vestibule into the IM vestibule. This translocation step apparently coincides with the rotation of the middle domains against the AAA domains, leading to a partial opening on the TM basket. Translocation into the IM vestibule then resulted in the delocalization of the TD domain, a prerequisite for the eventual release of the folded substrate into the IMS (Rip1-FeS) and the IM (Rip1-TM domain). Failure to hydrolyze ATPγS, however, leads to a —at least partial—locking of the substrate inside the IM vestibule as observed in our structure, thus leading us to speculate that ATP hydrolysis may be coupled with substrate release.

In addition to the matrix vestibule constriction, an additional driving force for the movement of Rip1-FeS toward the IM vestibule may be the electrostatic interaction between the partially negatively charged surface of the substrate and the positively charged residues that line the inner walls of the IM vestibule in Bcs1 (Fig. 4C). Indeed, one plausible model fitting our Rip1 density would position the N-terminus of Rip1 towards the matrix side exposing the negative charges towards the TD basket (Figs. 4C and EV5). Furthermore, several positively charged residues are conserved across different species (Appendix Fig. S5), amongst them R69 and R81. We thus investigated R69 and R81 mutants for respiratory growth defects in a non-fermentable carbon source as described above (Fig. 4D). We observed that R69A and R69E mutations lead to a lethal phenotype. Moreover, R81 mutations show a detectable growth defect. Of note, mutation of R81 in *H. sapiens* (R45C) is one of the mutations associated with the GRACILE syndrome (Al Qurashi et al, 2022), which highlights the role of selective pressure on maintaining a positively charged surface in this region of Bcs1.

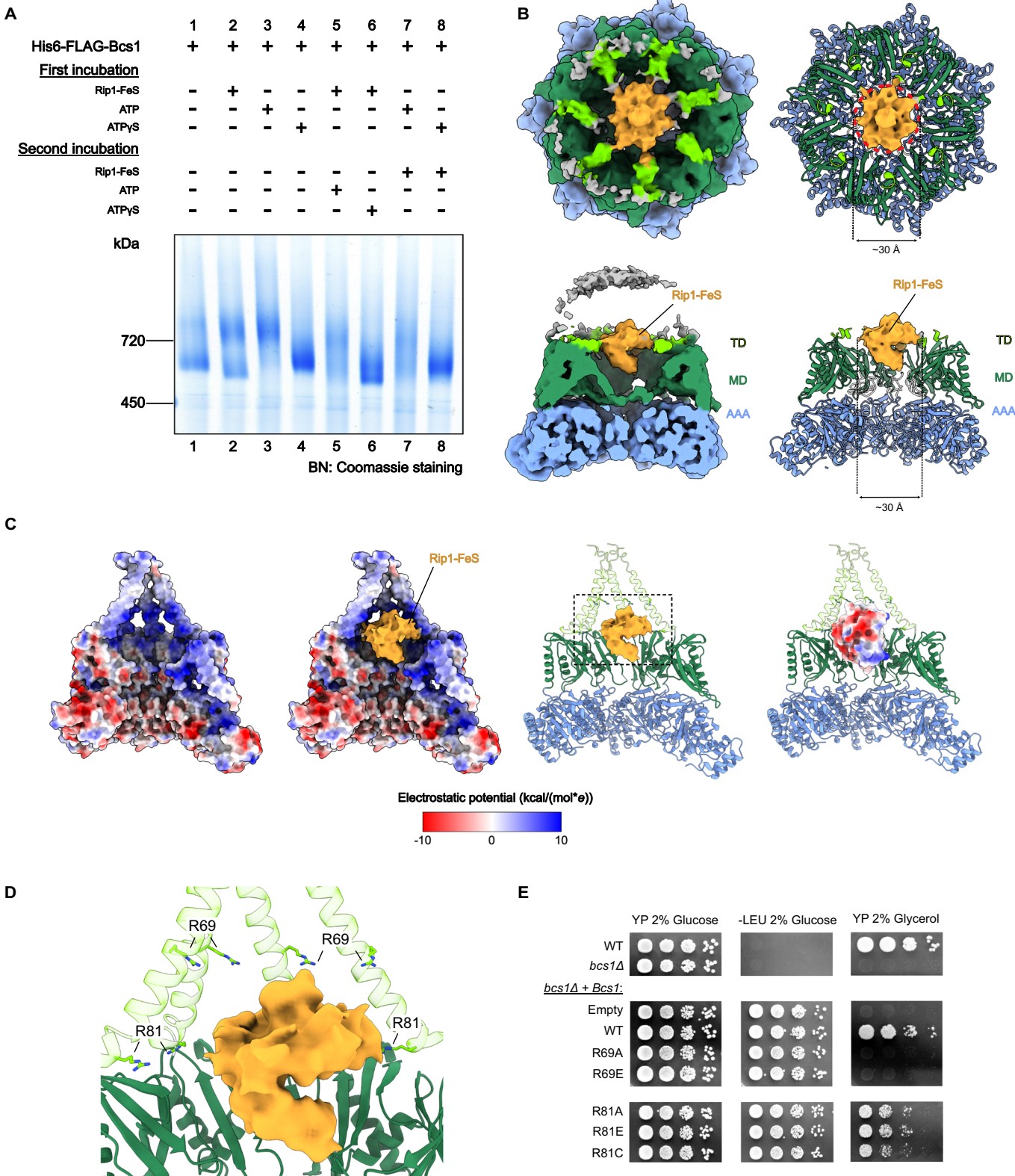

◀ **Figure 4. Structure of a Bcs1-Rip1 substrate gating intermediate in ATPγS2 state.**

(A) Blue native (BN) gel showing the in vitro reconstitution of the Bcs1-Rip1 complex in the presence of ATP and ATPγS. Purified Rip1-FeS was added to purified Bcs1 prior to (lanes 5, 6) or after (lanes 7, 8) the addition of the nucleotide. (B) Cryo-EM structure (left panels) and molecular model (right panels) of a Bcs1-ATPγS-Rip1-FeS gating complex. Upper panels show top views from IMS side, lower panels cut side views. The seal loops are shown transparent as in Fig. 3D. Approximate size of the Rip1-FeS density is indicated. (C) Cut side view of the electrostatic potential map of Bcs1-ATPγS2 (leftmost panel), Rip1-FeS density superimposed into the electrostatic potential map of Bcs1-ATPγS2 (left panel), Rip1-FeS density superimposed in the atomic model of Bcs1-ATPγS2 (right panel) and Rip1-FeS model docked in the atomic model of Bcs1-ATPγS2 (rightmost panel), displaying the overall charge distribution on the IM vestibule, substrate surface and substrate density. The relative position of the substrate in the central panel was approximated according to the extra density detected in the IM vestibule as shown in (B). A display of TD flexibility is omitted in the electrostatic map for better visualization of the overall location of the positively charged patch, but it is indicated in the atomic model, where it is shown as transparent. (D) Close-up view of enclosed area in C (right panel) displaying the position of conserved positively charged residues surrounding the substrate density. TD flexibility is indicated by the transparency of the helices. (E) Mutational analysis of the positively charged residues in Bcs1 TD based on a growth assay as described in Fig. 2. This panel is an excerpt of the comprehensive combined display shown in Appendix Fig. S6, reduced here to emphasize the residues discussed in the text and shown in (D). Source data are available online for this figure.

## Discussion

The translocation of a fully folded protein across the inner mitochondrial membrane requires a source of energy for directional movement across the hydrophobic barrier of the lipid bilayer and, at the same time, must at large preserve the membrane permeability barrier to maintain the proton gradient across the IM. We found how Bcs1-mediated Rieske translocation meets these requirements by employing an airlock-like mechanism that can be divided into three principal steps, loading of Bcs1 with the Rieske, gating of Rieske through the Bcs1 pore and release of both, the folded FeS domain into the IMS and the TM helix into the lipid bilayer of the IM (Fig. 5). In this work, we were able to visualize Rieske translocation intermediates, providing structural insights into the loading and gating steps of Rieske translocation.

Interestingly, we observe stable complex formation between Bcs1 and Rip1 only in the apo state (Apo1), i.e., with no nucleotide bound. Our structure of this complex clearly showed a substrate loading state with the Rip1-FeS stably accommodated within the Bcs1 matrix vestibule. A similar position was observed previously in the cryo-EM structure of the bovine Rieske FeS domain bound to the mammalian mBCS1 (Zhan et al, 2024). Unfortunately, however, the resolution of this structure did not allow for unambiguous positioning of the FeS domain, identification of molecular interactions or identification of the nucleotide state of mBCS1.

Our structure provides molecular details for the interaction between Rip1-FeS and Bcs1 and allows for the unambiguous positioning of the Rip1-FeS binding four Bcs1 protomers. In this conformation the C-terminus as well as the N-terminal region that would be following the TM domain, that were not visualized, are facing towards the seal pore of Bcs1. The Rip1-FeS interacts mainly via positively charged residues in Bcs1 and negatively charged patches on the Rip1 which form complementary surfaces. Here, we identified the DDR and EWR motifs in Bcs1 that interact with the substrate surface and showed that mutation of charged residues within those motifs was lethal in respiratory conditions, underlining their functional importance. In the observed position, the Rip1-FeS is transiently restricted in its orientation within Bcs1. This distinct mode of interaction may enable Bcs1 to probe the folding state of the substrate, explaining why mutants disturbing the characteristic antiparallel β-structure of the FeSD are deficient for translocation (Wagener et al, 2011). This is also in line with our observation, that reduction of the disulfide bond between C164 and C180, important for the overall fold of Rip1, results in a substrate

unable to bind Bcs1 (Fig. EV3B). The indirect probing of the presence of the 2Fe–2S cluster by Bcs1 may explain why it is not an absolute prerequisite for the translocation and IM insertion of Rip1 but required for its efficiency (Graham and Trumpower, 1991). Thus, the binding preference of Bcs1 to properly folded and 2Fe–2S cluster containing Rip1 over only partially folded Rip1 lacking the 2Fe–2S cluster appears to be established by the large interaction region in the matrix vestibule which is complementary to the fully folded Rip1.

Binding of folded Rip1 to the open matrix vestibule of Bcs1 is sterically only possible in the apo and ADP-bound states. Since we never observed Rip1 bound to Bcs1 in the ADP-bound state biochemically or by cryo-EM, we speculate that Rip1 either binds to the apo state directly or triggers the release of ADP when engaging Bcs1. This would be in agreement with the clearly different conformations of Bcs1 observed in the apo or ADP-bound state. This is different from the observations in the mammalian system where the mBCS1 loading complex (Zhan et al, 2024) adopted the ADP conformation. However, since mBCS1 ADP and Apo states were indistinguishable at the reported resolution of 7.2 Å it is unclear whether in the mammalian system the Rieske initially binds to the ADP state followed by ADP release, or whether ADP release is coupled to substrate release as suggested for mBCS1 (Zhan et al, 2024).

But how does ATP binding or ATP hydrolysis eventually trigger Rip1 translocation after its recruitment to the matrix vestibule? Here, we were able to determine the structures of Bcs1 in the presence of ATPγS with and without bound Rip1. We observed that upon nucleotide (ATP) binding, Bcs1 undergoes a substantial conformational change, displaying a dramatic narrowing of the matrix vestibule and concomitant opening of the seal pore. In the absence of Rip1, we found an equilibrium of two distinct ATPγS states, ATPγS1 and ATPγS2, which differ by a rotation of the Bcs1 MD accompanied with a (partial) delocalization of the TM basket forming the matrix vestibule. Interestingly, we could trap Rip1 in the ATPγS2 state after incubation of Bcs1 with Rip1 followed by the addition of ATPγS. In this structure, we find Rip1 translocated through the open seal pore into the IM vestibule, likely still clogging the pore from the IM side. Rip1 may be trapped in this position because, as described in a previous study (Wagener et al, 2011), release requires ATP hydrolysis and resetting of its conformation to the ADP state. Moreover, it was shown that TM deletion mutants, as used in this study, were retained in the Bcs1 complex in in vitro chase experiments in the presence of another slow-hydrolysable ATP analog, AMP-PNP. Notably, the Rip1-FeS

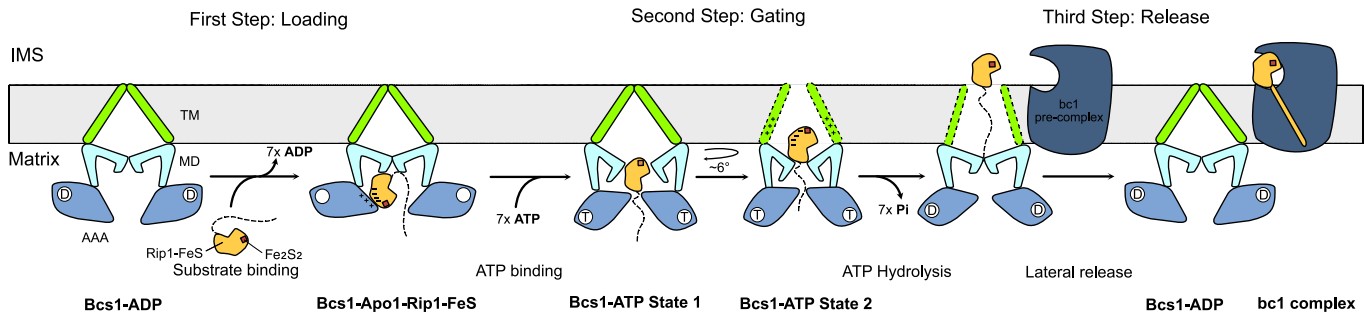

**Figure 5. Airlock mechanism of Bcs1-mediated Rip1 translocation across the IM.**

Schematic representation of the different states that Bcs1 goes through during Rip1 translocation. Dashed lines represent structural flexibility of transmembrane segments in Rip1 and Bcs1. Positive and negative charges are indicated in the Bcs1-Apo-Rip1-FeS and Bcs1-ATP State 2 as plus and minus signs.

is now located in the aqueous funnel of the IM vestibule clad by a number of positively charged residues that may again attract the negatively charged Rip1-FeS. Yet, in this state we do not observe any distinct interactions between Bcs1 and Rip1, indicating that the Rip1-Bcs1 interaction is not well defined but rather allows for multiple orientations of Rip1. Comparing the two structures of Rip1-bound Bcs1 in the substrate loading and gating states clearly suggests that ATP binding triggers a conformational change in Bcs1 that leads to unidirectional translocation of the Rip1-FeS from the matrix vestibule to the IM vestibule. Our structures thus confirm and refine the proposed airlock-like mechanism by which a concerted constriction of the matrix vestibule would push the Rip1-FeS towards the seal pore, which opens the gate into the IM vestibule. Charge complementarity between the positively charged surface of the IM vestibule of Bcs1 and the negatively charged surface of Rip1 may serve as an additional driving force and may also prevent the dissociation of the substrate back into the mitochondrial matrix upon ATP binding.

Based on these findings, we propose a refined model for Bcs1-mediated Rip1 translocation across the IM (Fig. 5). Our structure of the substrate loading state clearly shows that Bcs1 is in the nucleotide-free state when bound to the Rip1-FeS. In the next step ATP can now bind to the substrate loading complex, which leads to a conformational change in the entire Bcs1 heptamer. The dramatic constriction of the matrix vestibule drives translocation in the direction of the seal pore that opens to allow charge-driven passage of Rip1 to the IM vestibule. ATP binding results in at least two intermediates, here described as ATPγS1 and ATPγS2. We propose that the first intermediate, that we never observe with a bound Rip1, may occur only at an early stage of the translocation. In this very transient state, Rip1 may still be bound at or within the seal pore and the Bcs1 TM basket is closed. Rotation of the middle domain partially opens the TM basket allowing Rip1 to be accommodated in the IM vestibule, leading to a pre-release state. Moreover, our structures give a hint on how the permeability barrier during this translocation process may be maintained. Although contacts between translocated Rip1 and Bcs1 are loose and the resolution is thus rather poor, it appears plausible that the substrate itself is part of the permeability barrier by contributing to the closure of the seal pore. Another possibility is that the compacted AAA domains at the matrix side, as observed in the ATPγS states, also contribute to form a barrier. At least at low

contour levels, we observe in our density maps that the central pore to the matrix is closed, presumably by flexible loops. The final step to complete the translocation process is the release of the Rip1-FeS to the IMS and the integration of the Rip1-TM into the IM bilayer, eventually allowing integration into the bc1 complex.

This step has been shown to be ATP hydrolysis dependent (Wagener et al, 2011), and can be assumed to reset the conformation to the observed ADP state. We speculate that this conformational transition from the ATP to the ADP state may occur with the Bcs1 TMD basket still open towards the IMS, as observed in the ATPγS2 state. This would allow for release of the Rip1-FeS domain into the IMS and for the Rip1-TM to laterally integrate into the IM. It should be noted that for the final release of the Rip1-TM it would be sufficient for the basket helices of the TD of Bcs1 to laterally open and allow for partitioning of the Rip1-TM into the bilayer of the IM. Unfortunately, the visualization of the basket helices appears to be difficult and not only reflecting conformational differences but rather qualitative and quantitative aspects in the different datasets (Appendix Table S2).

Therefore, while our structures provide mechanistic details for the loading and gating step, several fundamental questions remain open with respect to the insertion and release of Rieske's TM by Bcs1. The native topology of all Rieske proteins is TM $N_{Matrix}$ – $C_{IMS}$, i.e., the N-terminus faces the matrix side while the C-terminal FeS is in the IMS. Notably, in our loading state, the Rip1-TM would be positioned facing towards the seal pore, but we cannot visualize it in our structure indicating that it is not (yet) ordered. Nevertheless, inversion of the globular FeS orientation relative to the pre-translocation state (Fig. 1D) would be required to adopt the native topology. This inversion may occur during the transition of the FeS through the seal pore. Notably, in other species like *M. musculus* and *H. sapiens*, the N-terminal region is even extended by a folded globular domain, representing another subunit of the bc1 complex, which remains in the matrix side of the membrane after insertion of Rip1. Consequently, in contrast to the situation in yeast, release of this N-terminus requires a complete lateral opening of the mBCS1 protomers, a step that was suggested to occur during ADP/ATP exchange for mBCS1 (Zhan et al, 2024). But even in the simpler situation in yeast, where the Rip1-TM just must laterally slip through the basket helices of Bcs1, it remains largely elusive at which exact stage of the ATPase cycle, and under which conditions the Rieske TM is released into the IM. It was shown before that

Bcs1 can interact with the bc1 complex (Zara et al, 2009) and although one study demonstrates that translocation and release of Rip1 are apparently independent of bc1 (Wagener et al, 2011), it is tempting to speculate that under physiological conditions Rieske TM release is coupled to the nearby presence of an assembly intermediate of the bc1 complex. Interestingly, it has recently been shown that folded Rieske interacts with an inner membrane-bound factor after Bcs1 translocation and insertion, during bc1 assembly (Zerbes et al, 2025). Eventually, ATP hydrolysis or release of ADP and exchange to ATP may be triggered or influenced by bc1 or another factor, yet so far experimental evidence for this hypothesis is lacking. In this context, we noted that in vitro Bcs1 permanently hydrolyzes ATP in the absence of substrate, which is highly unlikely to occur in vivo, further speaking for a contribution of the native membrane environment to Bcs1's ATPase cycle regulation. Thus, based on our structural work, we suggest that after Rip1-bound Bcs1 arrived in the pre-release state, ATP hydrolysis occurs in a conditional manner and leads to a transition state with the substrate still loosely bound to Bcs1. A direct handover and proper insertion into the bc1 pre-complex are then accompanied by Bcs1 dissociation and transition to the ADP state. Further experiments will be required to answer this question.

# Methods

**Reagents and tools table**

| Reagent/resource | Reference or source | Identifier or catalog number |
|---|---|---|
| **Experimental models** | | |
| *S. cerevisiae* W303 | Thomas and Rothstein, 1989 | |
| *S. cerevisiae* W303 *bcs1Δ*::natNT2 | This study | |
| *S. cerevisiae* W303 *rip1Δ*:: natNT2 | This study | |
| *E. coli* DH5α | NEB | C2987 |
| *E. coli* BL21 | Novagen | 70235-M |
| **Recombinant DNA** | | |
| pETDuet-TwinStrep-3C-Rip1$^{Δ30, Δ55-65}$ | This study | |
| pETDuet-TwinStrep-3C-Rip1$^{Δ81}$ | This study | |
| YEplac112-GAL-His$_6$-FLAG-Bcs1 | This study | |
| YCplac111-P.Rip1-Rip1$^{Δ30}$ | This study | |
| YCplac111-P.Bcs1-FLAG-Bcs1 | This study | |
| **Chemicals, enzymes, and other reagents** | | |
| Kaiser SC Single Drop-Out -LEU | Formedium | DSCK1004 |
| Phusion High Fidelity DNA Polymerase | NEB | M0531S |
| Digitonin | Calbiochem | 300410 |
| GDN (glyco-diosgenin) | Anatrace | GDN101 |
| His cOmplete Resin | Roche | 5893682001 |
| Strep-Tactin XT 4Flow | Iba | 2-5010-025 |
| Superose 6 Increase 10/300 GL | Cytiva | 29091596 |
| Superdex 200 Increase 10/300 GL | Cytiva | 28990944 |
| NativePAGE gels | Thermo Fisher | BN1003BOX |

| Reagent/resource | Reference or source | Identifier or catalog number |
|---|---|---|
| ATPγS | Jena Bioscience | NU-406-5 |
| **Software** | | |
| Cryosparc v4.4 | Punjani et al, 2017 | |
| Alphafold Database | http://alphafold.ebi.ac.uk | |
| ISOLDE | Croll, 2018 | |
| Coot | Emsley and Cowtan, 2004 | |
| ChimeraX v1.8 | Goddard et al, 2018 | |
| Phenix | Adams et al, 2010 | |
| ImageJ (Fiji) | Schindelin et al, 2012 | |
| GraphPAD Prism v10 | Dotmatics | |
| Affinity Designer 2 | Affinity | |
| **Other** | | |
| Titan Krios + Falcon 4i + SelectrisX Energy Filter | Thermo Fisher | |
| Vitrobot Mark IV | Thermo Fisher | |
| ÄKTA Pure | Cytiva | |
| Tecan Infinite M1000 | Tecan | |

## Plasmids, strains, and growth conditions

The coding region of the yBcs1 gene (YDR375C, residues 1-456) was cloned into a YEplac112 vector, downstream to a GAL bidirectional promotor and a N-terminal 6xHis-FLAG-3C tag. The plasmid was transformed into a *Saccharomyces cerevisiae* W303 (Thomas and Rothstein, 1989) *bcs1Δ* strain using the lithium acetate method. For the overexpression of yBcs1, a single colony isolated from a YPG plate (1% yeast extract, 2% peptone, 2% glucose, 2% Bacto-agar) was grown at 30 °C on synthetic complete media lacking leucine, containing 2% glucose (SDC-LEU-Glucose) to a final OD$_{600}$ of 1.0. The cells were then shifted to YPGal (1% yeast extract, 2% peptone, 2% galactose) and further grown until an OD$_{600}$ of ~4.0 was reached. Cells were harvested and flash frozen for storage at −80 °C.

The yRip1 gene without the mitochondrial targeting sequence (YEL024W, residues 30–215) was cloned into a pET-Duet plasmid after truncation of the mitochondrial targeting sequence and the full transmembrane helix coding region (residues 30–81) or only a hydrophobic segment (residues 55–64), downstream to a N-terminal TwinStrep-3C tag. For protein expression, the plasmids were transformed to an *Escherichia coli* BL21 strain. Single colonies isolated from LB-Amp plates (lysogeny-broth, 100 µg/mL ampicillin) were grown at 37 °C in LB-Amp liquid media to an OD$_{600}$ of 0.5–0.7, induced with IPTG to a final concentration of 0.5 mM, and supplemented with Fe$_2$(SO$_4$)$_3$ and L-Cysteine to a final concentration of 0.5 mM. The cells were further grown for 18 h at 16 °C. Cells were harvested and flash frozen for storage at −80 °C.

For the growth assays shown in Figs. 2D and 4E and Appendix Fig. S6A, the upstream UTR region (250 bp) of the yBcs1 gene was cloned to a YCplac111 vector (YCplac111-P.Bcs1). N-terminal

FLAG-tagged yBcs1 (1-456) was cloned downstream, and selected mutations were introduced by site-directed mutagenesis based on standard PCR techniques using Phusion polymerase (NEB). The mutations were confirmed by sequencing. The constructs were transformed as described above to the *bcs1Δ* strain. A single colony was grown in SDC-LEU-Glucose to an $OD_{600}$ of 1.5–2.0 and then diluted to 1.0 with fresh media. Serial 10-fold dilutions were spotted on YP plates containing either 2% glucose or 2% glycerol and on SDC-LEU-Glucose plates. The plates were incubated for 2–3 days at 30 °C, or at 37 °C where indicated.

For the growth assays shown in Fig. 2E and Appendix Fig. S6B, the upstream UTR region (322 bp) and the mitochondrial targeting sequence (1–30) of the yRip1 gene were cloned to a YCplac111 vector (YCplac111-P.Rip1). yRip1 (31–215) was cloned downstream and selected mutations were introduced by site-directed mutagenesis based on standard PCR techniques using Phusion polymerase (NEB). The mutations were confirmed by sequencing. The constructs were transformed as described above to a *rip1Δ* strain. A single colony was grown in SDC-LEU-Glucose to an $OD_{600}$ of 1.5–2.0 and then diluted to 1.0 with fresh media. Serial 10-fold dilutions were spotted on YP plates containing either 2% glucose or 2% glycerol and on SDC-LEU-Glucose plates. The plates were incubated for 2–3 days at 30 °C, or at 37 °C where indicated. Contrast enhancement of the digital images was carried out with ImageJ (Fiji) (Schindelin et al, 2012) only for visualization purposes (see Source Data for Original Images).

## Protein expression and purification

For purification of Bcs1, pelleted yeast cells were thawed and resuspended in lysis buffer (2 mL/g pellet; 0.65 M Sorbitol, 100 mM Tris pH 8.0, 5 mM EDTA pH 8.0, 5 mM aminocaproic acid, 0.2% BSA, 1 mM PMFS). Cells were lysed after two passages in a Cell Disruptor (Constant Systems Ltd.) at 2500 bar. Supernatants were clarified by centrifugation, 30 min at 3000 g. The membrane fraction was separated by ultracentrifugation, 1 h at 120,000 $\times g$. Crude membranes were washed once in SW buffer (0.65 M Sorbitol, 100 mM Tris pH 7.5, 5 mM aminocaproic acid, 1 mM PMFS) and centrifuged for further 30 min at 120,000 $\times g$. The pelleted membranes were then resuspended in SH buffer (0,65 M Sorbitol, 10 mM HEPES pH 7.5), aliquoted to an approximate total protein concentration of 10 mg/mL, and flash frozen for storage at −80 °C. For solubilization, membranes were resuspended in solubilization buffer (30 mM HEPES pH 7.5, 150 mM potassium acetate, 3% digitonin) to a final protein concentration of 5 mg/mL. After 1 h incubation at 4 °C, the insoluble fraction was separated by ultracentrifugation for 30 min at 120,000 $\times g$. The supernatants were incubated with pre-equilibrated (AP buffer: 30 mM HEPES pH 7.5, 150 mM KCl, 5% glycerol, 5 mM Imidazole, 0.1% digitonin) His cOmplete resin (Roche) for 2.5 h, 4 °C. They were subsequently washed with AP buffer supplemented with 50 mM imidazole and eluted with AP buffer supplemented with 250 mM imidazole. Selected fractions were pooled and concentrated in a 100 kDa cut-off Amicon (Merck). The concentrated sample was injected to a Superose 6 Increase 10/300 GL (Cytiva) column and run with SEC-GDN buffer (30 mM HEPES pH 7.5, 150 mM KCl, 5% glycerol, 0.024% glyco-diosgenin (GDN)). Final fractions were concentrated as described before and flash frozen or used immediately for downstream experiments.

For the purification of Rip1 constructs, frozen pellets were thawed and resuspended in lysis buffer (10 mL/g pellet; 100 mM Tris pH 8.0, 150 mM NaCl, 10% glycerol, 1 mM PMFS, 1x cOmplete protease inhibitor cocktail (Roche)). Cells were lysed after two passages in a Cell Disruptor (Constant Systems Ltd.) at 1250 bar. Supernatants were clarified by centrifugation, 15 min at 3000 $\times g$ followed by 25 min at 36,000 $\times g$. The supernatant was passed twice through a bed of pre-equilibrated (Strep-AP buffer: 25 mM HEPES pH 7.5, 150 mM NaCl, 10% glycerol) Strep-Tactin® XT 4Flow® (iba) resin. The resin was washed with Strep-AP, and the protein was eluted on the same buffer supplemented with 50 mM Biotin. The fractions were incubated with 3C-protease for 90 min and then concentrated in a 3 kDa cut-off Amicon for injection to a Superdex 200 Increase 10/300 GL (Cytiva) column, run with SEC buffer (30 mM HEPES pH 7.5, 150 mM KCl, 5% glycerol). Selected fractions were concentrated as before and flash frozen or used immediately.

## ATPase activity assay

Bcs1 ATPase activity was measured using the NADH (nicotinamide adenine dinucleotide)-coupled assay. Bcs1 (heptamer) in a final concentration of 65 nM (heptamer) was incubated in absence or in presence of 163 nM, 325 nM or 650 nM Rip1-TM substrate for 10 min at 25 °C in assay buffer (25 mM HEPES pH 7.5, 50 mM KCl, 2 mM $MgCl_2$, 0.024% GDN, 0.1 mg/mL BSA). After incubation, reactions were mixed with phosphoenolpyruvate 0.5 mM, NADH 0.1 mM, lactate dehydrogenase-pyruvate kinase (Merck) 25 U/mL and ATP 1 mM, to a final volume of 50 µL. Reactions were incubated at 30 °C, and fluorescence decay was monitored at 454 nm for 60 min in a Tecan Infinite M1000 plate reader. ATP concentration was calculated by interpolation in an ADP standard curve in the range 10–100 µM.

## BN-PAGE

Samples for BN-PAGE were prepared as follows. Bcs1 and Rip1 concentration was quantified using a Pierce™ BCA Protein Assay kit (Thermo Fisher Scientific). In total, 30 pmol of Bcs1 (heptamer) were diluted in BN-buffer (15 mM HEPES pH 7.5, 30 mM KCl, 5% glycerol, 0.1% digitonin) and incubated with varying quantities of Rip1 substrate (from 0 up to tenfold excess) in presence or absence of nucleotides and $MgCl_2$ (1 mM ADP, ATP or ATPγS, 2 mM $MgCl_2$). The final reaction volume was 30 µL. Depending on the experiment, the sample was incubated with substrate 10 min at 25 °C followed by the addition of the respective nucleotide and further incubated 10 min at 25 °C, or vice versa. The samples were mixed with NativePAGE™ Sample buffer (4X) and loaded to a NativePAGE™ gel (Thermo Fisher Scientific). Contrast enhancement of the digital images was carried out with ImageJ (Fiji) (Schindelin et al, 2012) only for visualization purposes (see Source Data for Original Images).

## Cryo-EM sample preparation and data collection

For the Bcs1-Apo-Rip1-FeS or Rip1-TM sample, 30 pmol of Bcs1 (heptamer) were incubated with a 10-fold excess of substrate for 20 min at 25 °C in BN-buffer. For the Bcs1-ATPγS sample, Bcs1 was incubated in BN-buffer supplemented with 2 mM ATPγS and

2.5 mM MgCl$_2$, for 20 min at 25 °C. For the Bcs1-ATPγS-Rip1-FeS sample, Bcs1 was first incubated with substrate in BN-buffer supplemented with 2.5 mM MgCl$_2$ for 10 min at 25 °C, followed by ATPγS addition to a final concentration of 2 mM and incubated for an additional 10 min. After incubation, all samples were kept on ice until plunge-freezing. A drop of 3.5 µL was applied onto glow-discharged Quantifoil Au 300 mesh R2/2 grids with an additional 3 nm layer of carbon. After incubation for 45 s, the grids were blotted for 3 s and plunge-frozen in liquid ethane, using a Vitrobot Mark IV (Thermo Fisher Scientific). Data collection was performed at 300 keV using a Titan Krios microscope equipped with a Falcon 4i direct electron detector and a SelectrisX Energy Filter (all Thermo Fisher Scientific) at a physical pixel size of 0.727 Å. Dose-fractioned movies were collected in a defocus range from −0.5 to 3.0 µm with a total dose of 60 e- per Å$^2$ and fractionated in 60 frames to obtain a total dose of 1 e- per Å$^2$ per frame.

## Data processing

Gain correction, movie alignment, and summation of movie frames were performed using MotionCor2 (Zheng et al, 2017). Further processing was carried out in CryoSPARC v4.4 (Punjani et al, 2017). CTF parameters for the micrographs were estimated using CTFFIND4 (Rohou and Grigorieff, 2015) (Bcs1-Apo-Rip1-FeS) or the Patch CTF Estimation job (Bcs1-Apo-Rip1-TM, Bcs1-ATPγS1/2, Bcs1-ATPγS2-Rip1-FeS).

### Bcs1-Apo-Rip1-FeS

Particle picking on 6044 micrographs was carried out by running a Blob Picker job, resulting in 800,580 particles. The particle set was extracted using a box size of 360 pixels, binned four times. 2D classification allowed the selection of a subset of 68,540 particles, which was used to train TOPAZ (Bepler et al, 2019). After TOPAZ particle picking, a set of 667,578 particles was selected and further cleaned by one round of 2D classification. In all, 546,208 particles belonging to good classes were used to generate an ab initio model, using the Ab-Initio Reconstruction job with three classes. After an initial round of heterogeneous refinement, the map with best features was employed to generate a consensus refinement of the entire particle set. Particles were then further classified in a second round of heterogeneous refinement, where a class with 320,760 particles displayed the best substrate density and was thereby selected for downstream refinement. Particles were re-extracted, two times binned, and sorted by unmasked 3D classification with 4 classes. In total, 309,698 particles from the best class were used in a focused 3D-classification job with three classes, where a circular mask covered the substrate density. A final subset of 106,497 particles was re-extracted and refined by a non-uniform refinement job, without imposing symmetry (C1), to a final resolution of 3.56 Å.

### Bcs1-Apo-Rip1-TM

A total of 1,415,540 particles were picked by a blob picker job on 13,310 micrographs. After extraction with a box size of 360 pixels, binned four times, the set was cleaned by successive 2D classification rounds, resulting in a subset of 119,984 particles, subsequently used for TOPAZ training. From TOPAZ picks, a total of 460,919 particles were cleaned with an additional 2D classification run and refined by a heterogeneous refinement job with 4

classes, using the Bcs1-Apo-Rip1-FeS map as reference. A class of 149,392 particles displaying the best resolved features was run in a second round of heterogeneous refinement. After the refinement, a set of 109,143 particles was re-extracted without binning, and a focused 3D classification with a mask in the substrate density was performed. A final subset of 57,257 particles was refined by a non-uniform refinement job (C1 symmetry) to a final resolution of 3.46 Å.

### Bcs1-ATPγS1/2

An initial set of 877,660 particles picked with Blob Picker from 8825 micrographs was used to generate templates for template picking. A set of 367,217 particles was obtained, extracted with a box size of 360 pixels, binned two times, and cleaned by successive rounds of 2D classification. 165,566 particles were selected, and a consensus map was obtained by homogeneous refinement. Multiple rounds of heterogeneous refinement with 4 classes were run to improve the quality of the map. A set of 234,802 particles belonging to a class with well-resolved features was obtained and refined by non-uniform refinement (C1 symmetry). 3D classification with two classes resulted in a class of 142,262 (61%, state S1) and a class of 92,540 (39%, state S2) differing by a rotation of 6° on the middle domain of Bcs1 and a worse resolved transmembrane region (state S2). The classes were re-extracted without binning and further refined by non-uniform refinement, enforcing C1 or C7 symmetry, to a final resolution of 2.92 Å (state S1–C1), 2.58 Å (state S1–C7), 3.09 Å (state S2–C1), and 2.74 Å (state S2–C7).

### Bcs1-ATPγS2-Rip1-FeS

From a subset of 2024 micrographs, 874,465 particles were picked using blob picker followed by template picker, extracted with a box size of 360 pixels, binned two times, and processed by successive rounds of 2D classification. 165,865 particles belonging to good classes were used to generate an ab initio model, using the Ab-Initio Reconstruction job. The particles were also used as templates for particle picking and 1,415,352 total particles were obtained from the full data set (18,385 micrographs). The map obtained was refined by non-uniform refinement (C1) and a heterogeneous refinement with three classes was performed to sort for overall structural heterogeneity. A class with the best structural features, containing 567,113 was re-extracted, two times binned and refined by non-uniform refinement. The refined map was used as input for a focused 3D-classification job, with a mask on the inner membrane vestibule. One class of 80,299 particles with strong density was selected. To confirm that the density was not an artifact from the processing methodology, the particles were input to an ab initio reconstruction job with two classes. This resulted in a junk class and a well-resolved class of 70,453 particles. These particles were re-extracted without binning and refined by non-uniform refinement (C1) to a final average resolution of 3.07 Å.

## Model building

A model of Bcs1 from the AlphaFold database (http://alphafold.ebi.ac.uk) was used to build the Bcs1-Apo-Rip1-FeS and Bcs1-Apo-Rip1-TM models. To model the substrate chain, the residues 82-215 of Rip1 from a model of the yeast cytochrome bc1 complex (PDB accession number 1KB9) were used. For the ATPγS models, an initial model was obtained using Modelangelo (Jamali

et al, 2024) on the Bcs1-ATPγS1 (C7) map. Real-space refinement was performed manually in COOT (Emsley and Cowtan, 2004). Multiple rounds of density fitting, and refinement were subsequently performed using ISOLDE (Croll, 2018) and real-space refinement in Phenix (Adams et al, 2010) . MolProbity (Chen et al, 2010) was used for model validation in Phenix. Docking and visualization of maps and models was done in ChimeraX (Goddard et al, 2018).

## Data availability

Cryo-EM density maps and atomic models have been deposited in the Electron Microscopy Data Bank (EMDB https://www.ebi.ac.uk/emdb/) and Protein Data Bank (PDB https://www.rcsb.org/) data bases under the accession codes EMD-51561 (Bcs1-Apo-Rip1-FeS), EMD-51537 and PDB:9GS2 (Bcs1-Apo-Rip1-TM), EMD-51552 and PDB:9GSN (Bcs1-ATPγS1 C7), EMD-53185 (Bcs1-ATPγS1 C1), EMD-51605 and PDB:9GU9 (Bcs1-ATPγS2 C7), EMD-53189 (Bcs1-ATPγS2 C1), and EMD-51562 (Bcs1-ATPγS2-Rip1-FeS).

The source data of this paper are collected in the following database record: biostudies:S-SCDT-10_1038-S44318-025-00459-4.

## Peer review information

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

## Acknowledgements

We thank Susanne Rieder and Charlotte Ungewickell for assistance with cryo-EM sample handling and data collection, Mariia Likhodeeva and Prof. Dr. Karl-Peter Hopfner (LMU, Gene Center Munich) for support with ATPase activity measurements, Dr. Dejana Mokranjac (LMU, BioCenter) and the members of the MitoClub for critical comments and suggestions, Dr. Birgitta Beatrix and Dr. Nives Ivić for their support with heterologous protein expression. This work was supported by funding from the German Research Foundation (Deutsche Forschungsgemeinschaft DFG; grant Nr. 510674444 to RB) and the European Research Council (ERC, Advanced Grant Nr. ADG 885711 to RB).

## Author contributions

**Cristian Rosales-Hernandez**: Conceptualization; Data curation; Formal analysis; Validation; Investigation; Visualization; Methodology; Writing—original draft; Writing—review and editing. **Matthias Thoms**: Methodology; Writing—review and editing. **Otto Berninghausen**: Data curation. **Thomas Becker**: Validation; Visualization; Methodology; Writing—original draft; Writing—review and editing. **Roland Beckmann**: Conceptualization; Supervision; Funding acquisition; Methodology; Writing—original draft; Project administration; Writing—review and editing.

Source data underlying figure panels in this paper may have individual authorship assigned. Where available, figure panel/source data authorship is listed in the following database record: biostudies:S-SCDT-10_1038-S44318-025-00459-4.

## Funding

## Disclosure and competing interests statement

The authors declare no competing interests.

# Expanded View Figures

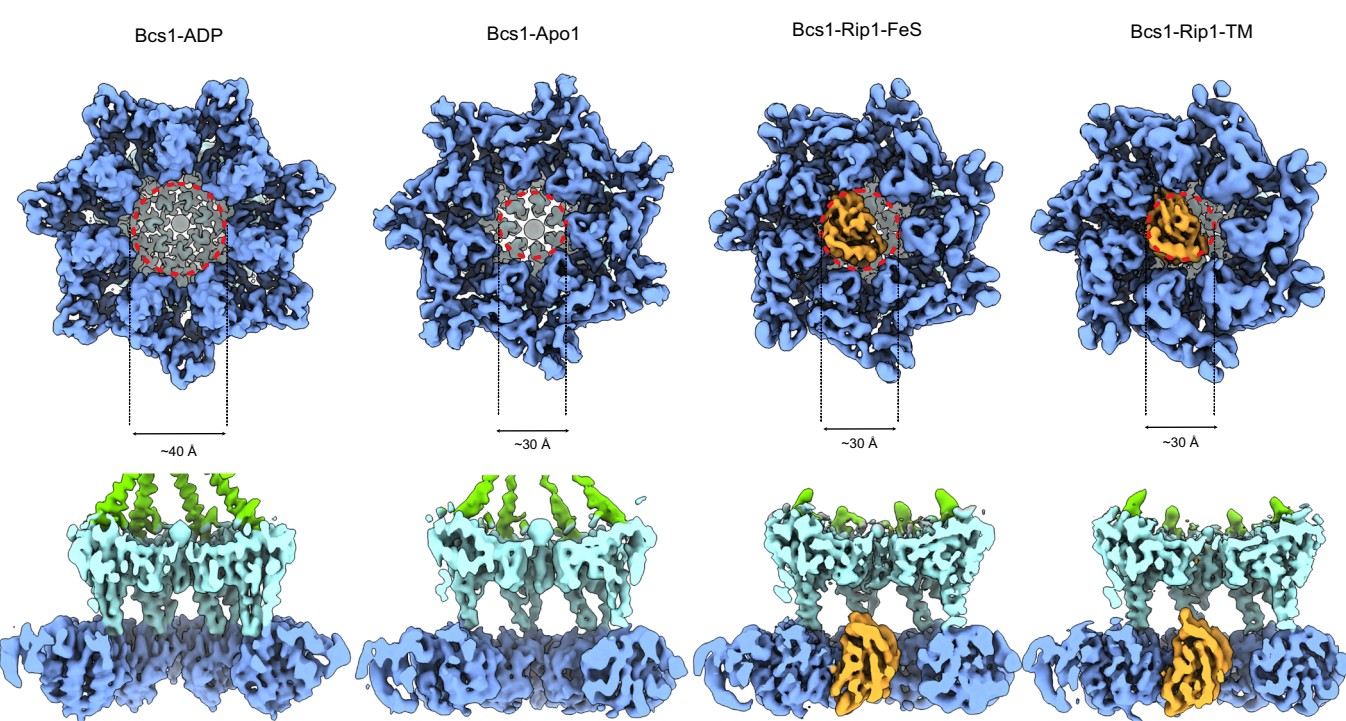

**Figure EV1.** **Comparison of yeast Bcs1 structures in ADP and Apo1 states.**

Cryo-EM structures shown at bottom (upper row) and side views (lower row) of Bcs1 in ADP state, Apo1 state (Kater et al, 2020) and bound to Rip1-FeS and Rip1-TM. Note that the Rip1-bound conformation of Bcs1 is Apo1. Red circles represent the diameter of the matrix vestibule.

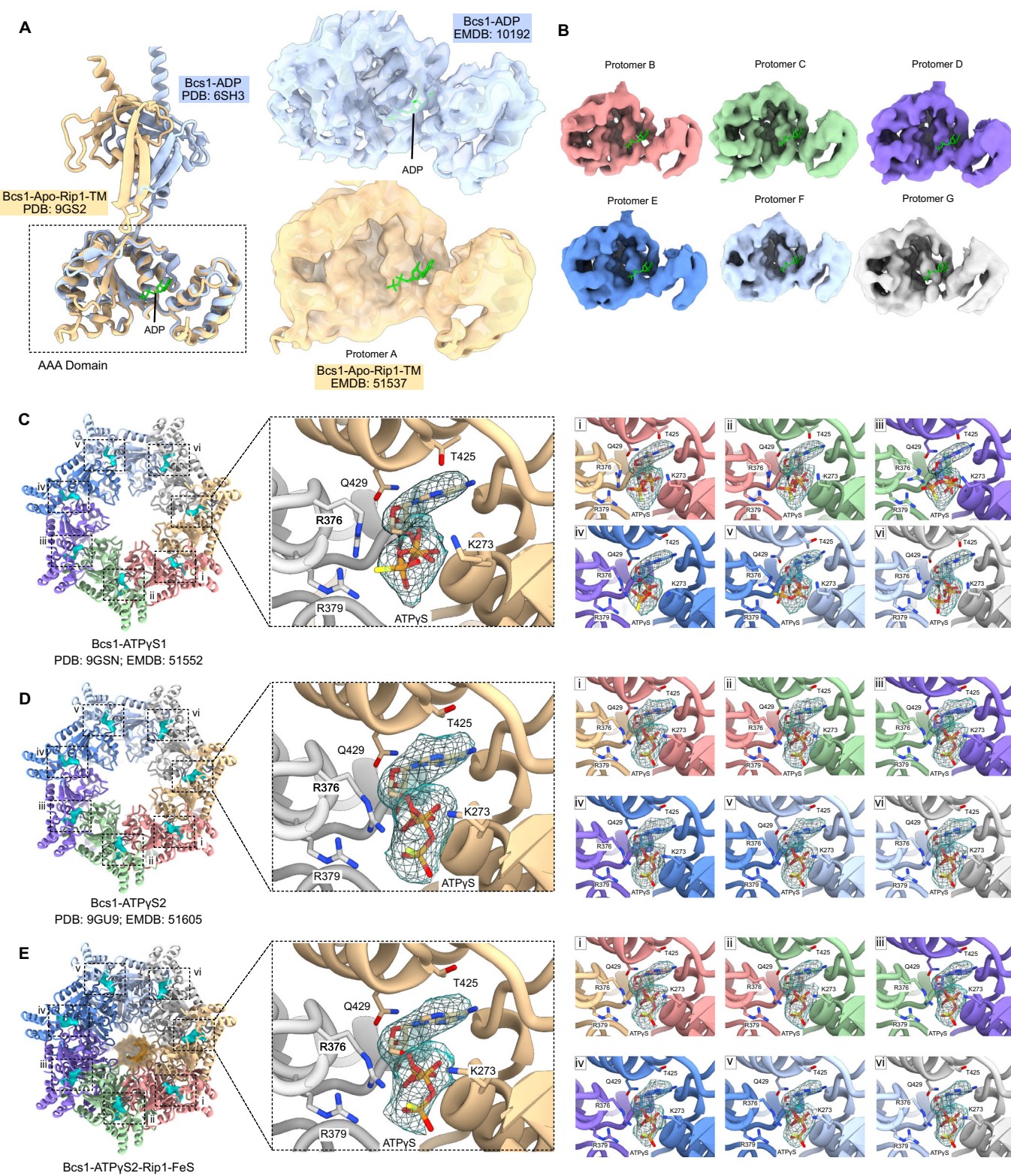

◀ **Figure EV2. ATP-binding pocket occupancy in the different Bcs1 states.**

(A) Left panel: Alignment of the atomic models of Bcs1-ADP (PDB:6SH3, chain A) and Bcs1-Apo1-Rip1-TM (PDB:9GS2, chain A), displaying the relative position of ADP in the binding pocket. Right panel: Close-up view on the AAA domain of protomer A of Bcs1 and map superposition, highlighting the density of ADP found in the Bcs1-ADP state (top) and absent in Bcs1-Apo1-Rip1-TM (bottom). (B) ADP nucleotide model superimposed on the AAA domain density of the protomers B-G of Bcs1-Apo1-Rip1-TM. (C) Left panel: Bottom view of Bcs1- ATPγS1 and isolated ATPγS density. Highlighted in squares are a close-up view of the density in the protomer A and, in boxes i-vi, ATPγS densities for protomers B-G. Right panel: Close-up views of the enclosed regions i-vi. (D) same as in C for Bcs1- ATPγS2. (E) Same as in (C) for Bcs1- ATPγS2-Rip1-FeS.

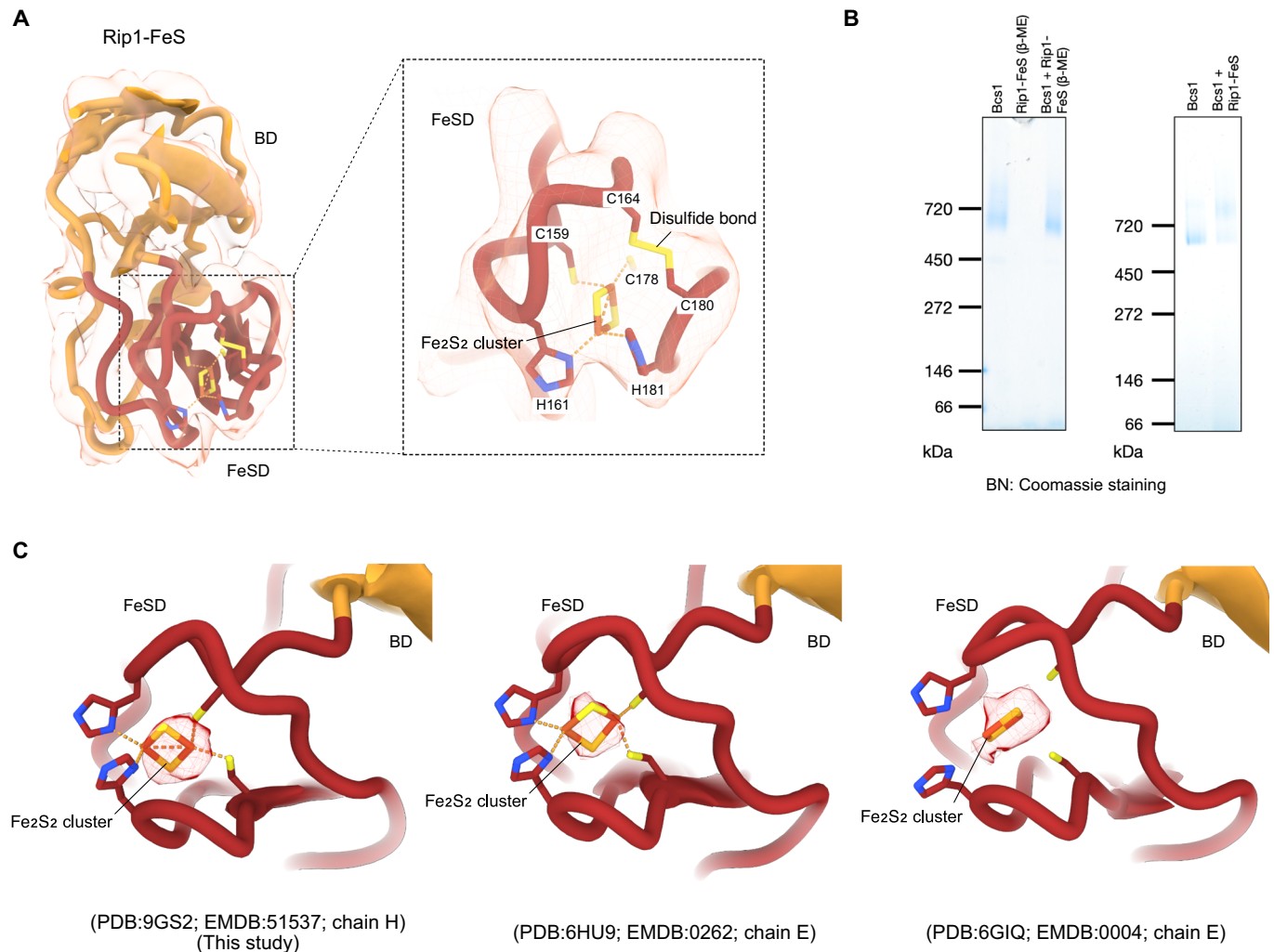

**Figure EV3.  Structural details of Rip1-FeS bound to Bcs1 in the Apo1 state.**

(**A**) Left panel: Rip1-FeS atomic model fitted into density, highlighting the FeSD domain. Right panel: Close-up view on the FeSD displaying the residues involved in coordination of the 2Fe–2S and the disulfide bond between C164 and C180. The 2Fe–2S cluster is shown in a stick representation, where yellow sticks represent the sulphur atoms and red sticks represent the iron atoms. (**B**) Blue Native (BN) gel showing the effect of β-mercaptoethanol (β-ME) on the binding of purified Rip1-FeS to Bcs1 (left) compared to Rip1-FeS purified in the absence of β-ME. (**C**) Comparison of the Rip1-associated 2Fe–2S cluster densities from Bcs1-Apo1-Rip1-TM and active CIII complex Cryo-EM maps. Source data are available online for this figure.

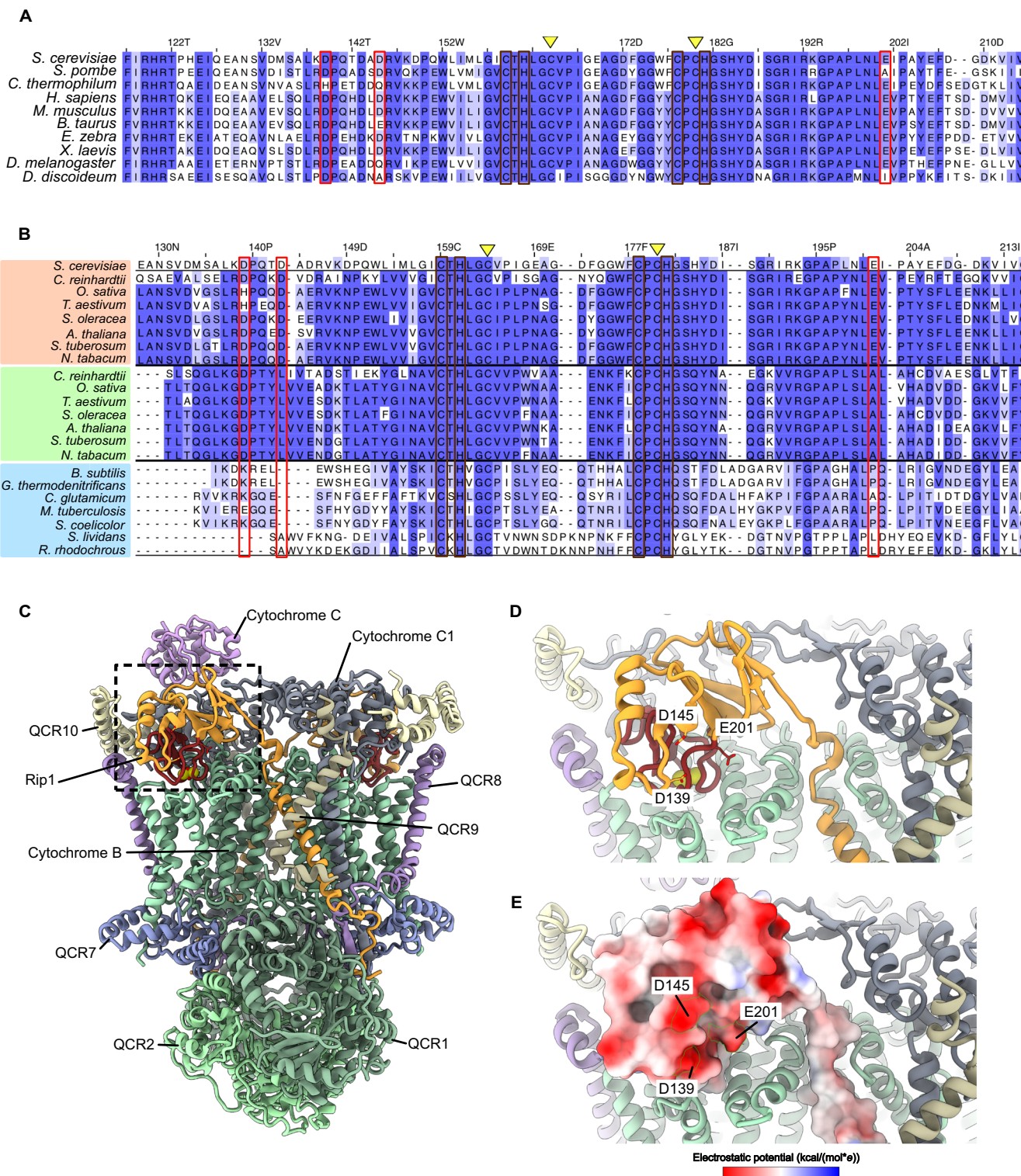

◀  **Figure EV4.  Conserved residues on the Bcs1-interaction surface of Rip1.**

(A) Multiple sequence alignment of the globular domain of Rip1 from selected species generated by Clustal Omega (Madeira et al, 2024) and displayed using Jalview (Waterhouse et al, 2009). Enclosed in red boxes are some of the negatively charged residues (D139, D145, E201) that interact with Bcs1 during translocation, in brown boxes are the residues responsible for the coordination of the 2Fe–2S cluster (C159, H161, C178, H181) and yellow arrows point to the residues that form a structurally relevant disulfide bond (C164, C180). The amino acids are colored according to their percentage of identity. (B) Multiple sequence alignment as in (A) but comparing the mitochondrial (pink box) Rip1 homolog of photosynthetic species with its chloroplast homolog petC in the same species (green box) and its prokaryotic homolog qcrA from prokaryote species (light blue box). (C) Structure of the fully assembled dimeric bc1 complex (PDB: 1KYO), highlighting the position of the globular domain of one of the two Rip1 subunits. (D) Close-up view on the enclosed region from (C), displaying Rip1 negatively charged amino acids. (E) Same view as in (D) but displaying the negatively charged electrostatic surface. The 2Fe–2S cluster is shown in a sphere representation, where yellow spheres represent the sulfur atoms and red spheres represent the iron atoms.

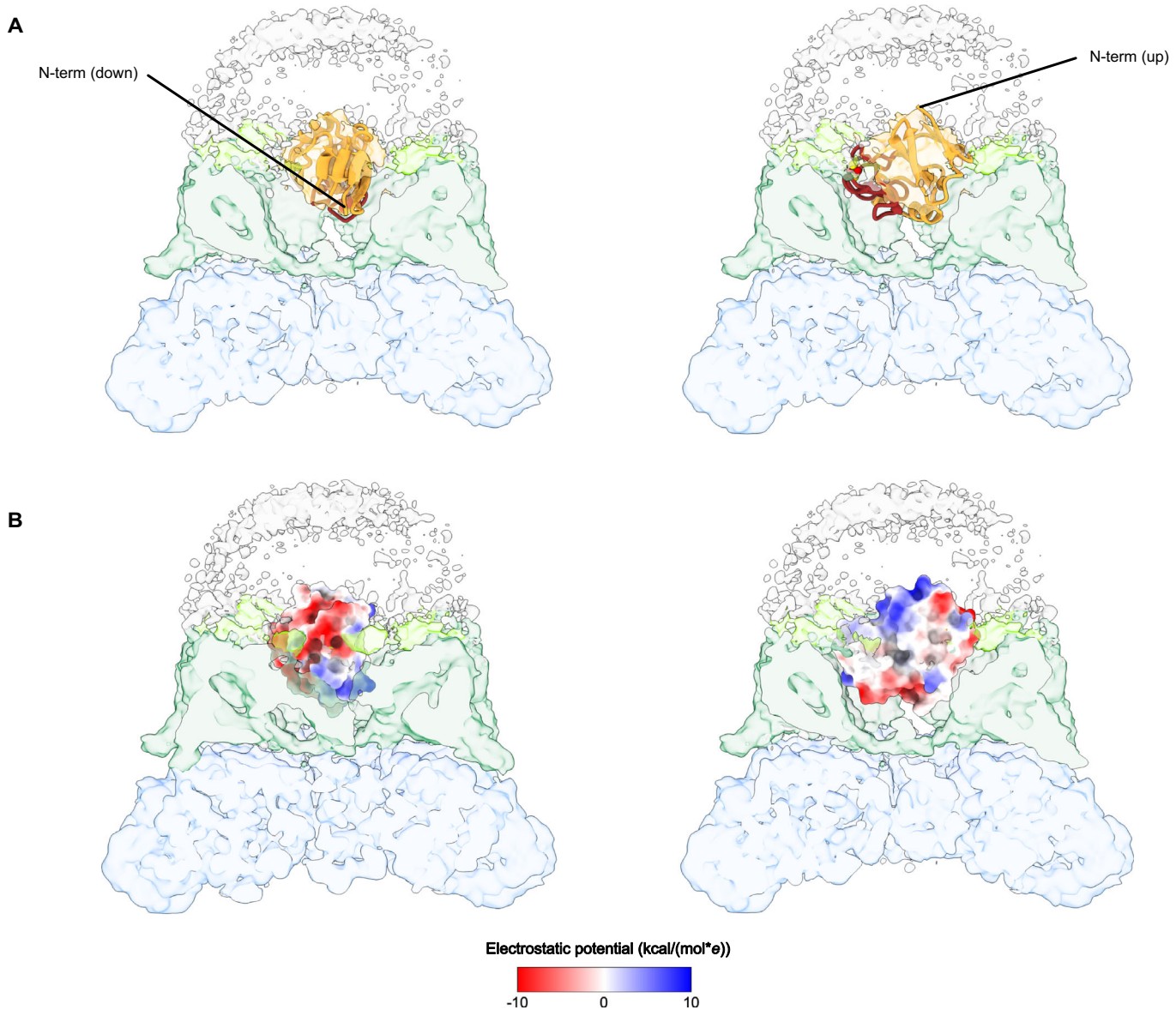

**Figure EV5. Plausible orientations of Rip1-FeS in the translocation state.**

(A) Cut side view of the atomic model of Rip1-FeS docked into the Bcs1-ATPγS2-Rip1-FeS map in an orientation that locates the N-terminal residues towards the mitochondrial matrix (left) or towards the IMS (right). The 2Fe–2S cluster is shown in a sphere representation, where yellow spheres represent the sulfur atoms and red spheres represent the iron atoms. (B) Electrostatic surface representation of the models displayed in (A).

