## [Peer Review File · The EMBO Journal]

Mechanistic insights into Bcs1-mediated mitochondrial membrane translocation of the folded Rieske protein

Cristian Rosales-Hernandez, Matthias Thoms, Otto Berninghausen, Thomas Becker, and Roland Beckmann

Corresponding author(s): Roland Beckmann (beckmann@genzentrum.lmu.de)

Review Timeline:

Submission Date:	13th Dec 24
Editorial Decision:	6th Feb 25
Revision Received:	9th Apr 25
Editorial Decision:	22nd Apr 25
Revision Received:	25th Apr 25
Accepted:	25th Apr 25

Editor: Hartmut Vodermaier

Transaction Report:

Prof. Roland Beckmann
Gene Center Munich, Department of Biochemistry, University of Munich, Germany
Feodor-Lynen-Str. 25
Munich, Bavaria 81377
Germany

6th Feb 2025

Re: EMBOJ-2024-119918
Mechanistic insights into Bcs1-mediated translocation of the folded Rieske protein

Dear Roland,

Thank you again for submitting your structural study on Bcs1-mediated Rieske protein translocation for our consideration, and apologies for the delay with its review, caused by limited referee availability around the turn of the years. We have now received a complete set of comments from three experts in AAA-ATPase structure and mitochondrial protein translocation, copied below for your information. All referees appreciate the potential importance of your results, as well as the experimental quality of the work, and would be supportive of publication pending satisfactory modification in response to a number of specific issues noted in their reports. As you will see, the majority of these concerns refer to presentation, interpretation, and deepened analysis of the obtained data; with only referee 1 also asking for some additional mutagenesis experiments.

Should you be able to adequately address the various points raised by the reviewers, we would be happy to pursue a revised version further for EMBO Journal publication. Please be reminded that it is our policy to allow only a single round of major revision, making it important to carefully respond to all points at the time of resubmission. Also, please do not hesitate to contact me already during the early stages of the revision work, in case you would like discuss how to best tackle particular issues. Finally, should you require more time than our default three-months revision period, we would be happy to offer an extension, during which our 'scooping protection' (meaning that competing work appearing elsewhere in the meantime will not affect our considerations of your study) would of course remain valid.

Further information on preparing and uploading a revised manuscript can be found below and in our Guide to Authors. Thank you again for the opportunity to consider this work for The EMBO Journal, and I look forward to your revision.

With kind regards,

Hartmut

- 4) Each main and each Expanded View (EV) figure should be uploaded as individual production-quality files (preferably in .eps, .tif, .jpg formats). For suggestions on figure preparation/layout, please refer to our Figure Preparation Guidelines: <http://bit.ly/EMBOPressFigurePreparationGuideline>
- 5) Point-by-point response letters should include the original referee comments in full together with your detailed responses to them (and to specific editor requests if applicable), and also be uploaded as editable (e.g., .docx) text files.
- 6) Please complete our Author Checklist, and make sure that information entered into the checklist is also reflected in the manuscript; the checklist will be available to readers as part of the Review Process File. A download link is found at the top of our Guide to Authors: embopress.org/page/journal/14602075/authorguide
- 7) All authors listed as (co-)corresponding need to deposit, in their respective author profiles in our submission system, a unique ORCID identifier linked to their name. Please see our Guide to Authors for detailed instructions.
- 8) Please note that supplementary information at EMBO Press has been superseded by the 'Expanded View' for inclusion of additional figures, tables, movies or datasets; with up to five EV Figures being typeset and directly accessible in the HTML version of the article. For details and guidance, please refer to: embopress.org/page/journal/14602075/authorguide#expandedview
- 9) To facilitate reproducibility and cross-laboratory adoption of methodologies, please structure the Materials & Methods section as outlined in our guide to authors, including a completed Reagents and Tools Table that can be downloaded from our author guidelines as well (<https://www.embopress.org/page/journal/14602075/authorguide#structuredmethods>).
- 10) Digital image enhancement is acceptable practice, as long as it accurately represents the original data and conforms to community standards. If a figure has been subjected to significant electronic manipulation, this must be clearly noted in the figure legend and/or the 'Materials and Methods' section. The editors reserve the right to request original versions of figures and the original images that were used to assemble the figure. Finally, we generally encourage uploading of numerical as well as gel/blot image source data; for details see: embopress.org/page/journal/14602075/authorguide#sourcedata

At EMBO Press, we ask authors to provide source data for the main manuscript figures. Our source data coordinator will contact you to discuss which figure panels we would need source data for and will also provide you with helpful tips on how to upload and organize the files.

In the interest of ensuring the conceptual advance provided by the work, we recommend submitting a revision within 3 months (7th May 2025). Please discuss the revision progress ahead of this time with the editor if you require more time to complete the revisions. Use the link below to submit your revision:

Link Not Available

Referee #1:

The manuscript by Rosales-Hernandez et al. reports the structure of the mitochondrial Bcs1 AAA+ ATPase bound to its Rieske substrate (Rip1). Bcs1 is an unusual member of the AAA+ ATPase family, assembling as a heptameric complex rather than the canonical hexameric rings. Its only known substrate, Rip1, is an iron-sulfur cluster-containing protein that is a component of mitochondrial Complex III. Bcs1 plays a crucial role in Complex III assembly by facilitating the translocation of Rip1 from the mitochondrial matrix to the inner membrane. The molecular mechanism by which Bcs1 accomplishes this process has remained elusive.

Previous studies have established that Bcs1 forms heptameric rings, and the prevailing model proposes that Rip1 translocation occurs through a concerted mechanism in which all seven subunits undergo a synchronous conformational change, enabling the folded substrate to move from the matrix to the inner membrane (Tang et al., 2020; Kater et al., 2020; Pan et al., 2023; Zhan et al., 2024). This mechanism contrasts sharply with the more widely accepted "hand-over-hand" model for hexameric AAA+ ATPases that translocate and unfold their substrates. However, the mechanistic details of this concerted model have remained poorly understood, largely due to the absence of structural data capturing Rip1 in complex with Bcs1.

The present study provides key structural insights by reporting Bcs1 in different nucleotide-binding states and, most importantly, in complex with Rip1. These structures reveal that Rip1 is loaded into the matrix vestibule through specific protein-protein interactions, which the authors validate through mutagenesis experiments. Another key structure captures Rip1 translocated toward the inner membrane, appearing poised for transfer to Complex III. Overall, this study represents an elegant and significant advance in our understanding of Bcs1 function and Complex III assembly. I support publication with the following suggestions:

1. The network of interactions between Rip1 and four protomers of Bcs1 in the matrix vestibule appears compelling, but the current surface charge representation of Bcs1 lacks sufficient detail to effectively illustrate these interactions. I suggest revising the figures to depict side-chain interactions more explicitly.
2. While the mutagenesis experiments on Bcs1 establish the importance of these residues, it would be valuable to determine whether mutations in Rip1 could produce similar lethal phenotypes.
3. The authors should clarify why the TM helices adopt either defined or less ordered conformations across different states. Is there a coordinated relationship between TM helix movement and other domains of Bcs1?
4. The presentation of Fig. 4C is potentially misleading, as it suggests that the TM helices are ordered in this state. However, this appears inconsistent with the text and with Fig. 4B, which shows a lack of ordered TM density.
5. The authors conclude that the translocated Rip1 within the inner membrane vestibule is less ordered, precluding model fitting. However, the density appears to have a characteristic crescent shape (side views in Fig. 4B & C). Would this shape preclude the ability to rigid-body fit a Rip1 model into the density?
6. Given the poor local resolution of translocated Rip1, further image processing might enhance the structural details. If particle alignment is driven by the matrix side of the complex, would focused refinement around the middle domains improve local resolution in this region?
7. It would be interesting to examine whether the Bcs1-Rip1 interactions are conserved across species. The authors provide an alignment of Bcs1, highlighting the EWR and DDR motifs (Fig. EV10), but do the interacting residues in Rip1 exhibit similar conservation? Investigating this could provide co-evolutionary insights into why Rip1 remains the sole substrate of Bcs1.

Other Comments:

1. The authors should clarify whether the ~30 Å diameter opening in the middle domains in ATP_γS-bound states is sufficient to accommodate Rip1 translocation.
2. There are several typographical errors throughout the manuscript, including:
"Bsc1" (page 9, heading)
"dfference" (page 12)
"postion" (page 14)
"disitinct" (page 15)
"concommittent" (page 15)

Overall, this is a well-executed study that significantly advances our understanding of Bcs1 function. I look forward to seeing the authors' revisions.

Referee #2:

This manuscript by the Beckmann group addresses an exceptionally fascinating conceptual problem in cellular biochemistry, the translocation of folded macromolecules across biological membranes. Studies on bacterial and peroxisomal systems as well as the nuclear pore have revealed that intrinsically disordered protein domains within translocator complexes represent an important feature of the passageway for folded proteins across membranes. Yet another, highly conserved and crucial factor of the mitochondrial inner membrane, the Rieske-Fe/S protein (Rip1), which requires co-factor binding prior to the final export step from the mitochondrial matrix, was shown to depend on a non-canonical AAA ATPase, termed Bcs1, for translocation. Despite many studies on the molecular mechanisms of Rip1 biogenesis, the mechanism of Bcs1-mediated Rip1 translocation has remained enigmatic.

Beckmann and colleagues show here for the first time cryo-EM images of the Rip1-loaded homo-heptameric Bcs1 complex at distinct stages of translocation in an excellent resolution that allows the visualization of molecular details and translocase-substrate interactions. The comparison of a substrate-docked pre-translocation state in the absence of nucleotides is thoroughly compared to advanced translocation intermediates formed in the presence of non-hydrolysable ATP analogues. In this way, the transition from substrate binding via the matrix-localized ATPase domain to occlusion in a Bcs1-specific middle domain and further to a localization within the membrane-embedded domain can be beautifully recapitulated.

The quality of the data is impressive and the manuscript is very well written, peppered with sophisticated details, but also accessible for non-expert readers. It certainly represents a major conceptual advance in an important area of cellular biochemistry and is well suited for publication in the EMBO Journal. However, the following two major points should be convincingly addressed by the authors prior to acceptance:

1. In mitochondria, insertion of the Fe-S cluster occurs before Bcs1-mediated translocation of Rip1 and co-factor bound Rip1 is the clearly preferred substrate for Bcs1, which is also acknowledged by the authors in the discussion. They even speculate, how co-factor incorporation may optimize substrate interaction with the ATPase domain. Given the advanced possibilities of molecular modeling, the authors are strongly encouraged to try and model a Fe-S cluster-bound holo-state of Rip1 into their structures of Rip1-Bcs1 binding and translocation complexes to test their hypothesis.

2. When discussing the driving forces for Rip1 to move on from early to later translocation stages, besides conformational changes in Bcs1, the authors suggest a role for positively charged amino acid residues (R69, R81) in the IM vestibule that may attract Rip1. The initial interaction between Rip1 and Bcs1 on the matrix site of the membrane in the pre-translocation complex is also suggested to be mediated by electrostatic interactions. Does the proposed transition make sense from the energetic perspective? The authors should analyze Bcs1 variants R69 and R81 with suitable amino acid substitutions. As altered substrate interactions and/or conformational state appear to affect the migration behavior of Rip1-Bcs1 complexes in native gels, this may offer a possibility to examine in vitro, if R69 and R81 are important for such transitions. At least, the authors should provide in vivo growth test for the respective bcs1 mutants as shown in Figure 2 to substantiate the role of the proposed docking site amino acid cluster

Minor points:

- I may have overlooked this, but what does the yellow ball within the Rip1 FeS domain that is visible in many structural models actually represent?
- Out of curiosity: A study by the Trumpower group has identified a structurally important disulfide bond in folded Rip1 between C164 and C180. Is this disulfide present in the structures of Rip1 translocation intermediates shown here?
- In the introduction the authors state that Rip1 incorporation into complex III is a pre-requisite for dimerization and super-complex formation and cite a paper from the year 2000. This is not anymore the current state of knowledge. Several groups have shown dimer formation of complex III and even super-complex formation (although inefficient) of intermediates lacking Rip1.

Referee #3:

Bcs1, an AAA-ATPase in the mitochondrial inner membrane (IM), is a unique translocator that mediates the transfer of a ligand-bound folded protein, Rieske FeS protein (Rip1 in yeast), from the matrix to the intermembrane space (IMS) across the IM without its unfolding. Although high-resolution cryo-EM structures of substrate-free Bcs1 and a low-resolution structure of Bcs1 bound to a substrate (Zhan et al., 2024) are available, the precise mechanism by which folded Rip1 is translocated through Bcs1 across the IM still remains elusive. In this manuscript, the authors purified full-length Bcs1 from yeast and Rip1 derivatives (Rip1-FeS and Rip1-TM) from *E. coli* and optimized conditions to reconstitute yeast Bcs1-Rip1 derivative complexes. They determined high-resolution cryo-EM structures of Bcs1(Apo)-Rip1, Bcs1(ATPγS), and Bcs1(ATPγS)-Rip1, and on the basis of these structures, they proposed a model for substrate transfer from the matrix toward IMS via Bcs1. In this model, Rip1 binds to nucleotide-free Bcs1 (loading step). Then, ATP binding to Bcs1 induces a conformational change, a constricting the matrix vestibule and simultaneously opening the seal pore between the matrix and IM vestibules, leading to substrate translocation to the IM vestibule (gating step). The reported structures, especially those of Bcs1 bound to a substrate are novel, and combined with the analysis of Bcs1-Rip1 complex formation, these findings significantly advance our understanding of how Bcs1 translocates a folded substrate across the IM without ion leakage, making this study of broad interest to NSMB readers. However, I have the following concerns for the authors to consider.

- Based on in vitro reconstitution under different nucleotide conditions, the authors suggest that Rip1 binds to nucleotide-free Bcs1. However, given the high cellular ATP concentrations, I am skeptical about the presence of a significant fraction of nucleotide-free Bcs1 waiting for a substrate. While Fig. 4A demonstrates efficient Bcs1-Rip1 complex formation, this does not necessarily mean that ATP-bound Bcs1 cannot accommodate Rip1 under conditions (e.g., varying ATP concentrations).
- Zhan et al (2024) showed that both the ADP-bound and nucleotide-free forms of mouse Bcs1 can form a complex with Rieske FeS protein. Does this discrepancy simply reflect a species difference?
- In Fig. 1A, is the lower band on the BN gel a Bcs1 monomer?
- In Fig. 1C, how reliable is the TD part in the Bcs1(Apo)heptamer-Rip1 complex? Does this form indeed lack nucleotides? It is important to show the EM density map around the nucleotide binding sites to confirm the absence of residual ADP retained during purification. Alternatively, residual nucleotides, if present, can be detected by UV measurements after heat denaturation of the Bcs1 sample used for EM measurements.
- In Fig. 2, the authors should describe the interactions between Bcs1 and Rip1 through conserved motifs and assess their importance by mutagenesis. Why were conserved residues W213 and D300 not discussed? In the central panel in Fig. 2C, should R212 be R214?
- In Figs. 3 and 4, ATPγS density should be shown to confirm the presence of the Bcs1(ATPγS) form.
- In Fig. EV1, how was the binding of Fe-S to the purified Rip1 verified?
- Page 7, line 26: 79.156 is 106,497?
- Page 7, line 27: 57.257 is 57,257?
- Page 9, line 17: Bsc1 should read Bcs1.

- Page 12 (for Fig. 4A): the BN-gel lanes should be labeled clearly and referred to by lane numbers in the text for better readability.
- Page 16, line 10: analogon is analog?
- Page 22, line 2: Increase 10/30 GL is Increase 10/300 GL?

Referee #1:

The manuscript by Rosales-Hernandez et al. reports the structure of the mitochondrial Bcs1 AAA+ ATPase bound to its Rieske substrate (Rip1). Bcs1 is an unusual member of the AAA+ ATPase family, assembling as a heptameric complex rather than the canonical hexameric rings. Its only known substrate, Rip1, is an iron-sulfur cluster-containing protein that is a component of mitochondrial Complex III. Bcs1 plays a crucial role in Complex III assembly by facilitating the translocation of Rip1 from the mitochondrial matrix to the inner membrane. The molecular mechanism by which Bcs1 accomplishes this process has remained elusive.

Previous studies have established that Bcs1 forms heptameric rings, and the prevailing model proposes that Rip1 translocation occurs through a concerted mechanism in which all seven subunits undergo a synchronous conformational change, enabling the folded substrate to move from the matrix to the inner membrane (Tang et al., 2020; Kater et al., 2020; Pan et al., 2023; Zhan et al., 2024). This mechanism contrasts sharply with the more widely accepted "hand-over-hand" model for hexameric AAA+ ATPases that translocate and unfold their substrates. However, the mechanistic details of this concerted model have remained poorly understood, largely due to the absence of structural data capturing Rip1 in complex with Bcs1.

The present study provides key structural insights by reporting Bcs1 in different nucleotide-binding states and, most importantly, in complex with Rip1. These structures reveal that Rip1 is loaded into the matrix vestibule through specific protein-protein interactions, which the authors validate through mutagenesis experiments. Another key structure captures Rip1 translocated toward the inner membrane, appearing poised for transfer to Complex III. Overall, this study represents an elegant and significant advance in our understanding of Bcs1 function and Complex III assembly. I support publication with the following suggestions:

We thank the reviewer for the positive evaluation of our work and recommendation for publication.

1. The network of interactions between Rip1 and four protomers of Bcs1 in the matrix vestibule appears compelling, but the current surface charge representation of Bcs1 lacks sufficient detail to effectively illustrate these interactions. I suggest revising the figures to depict side-chain interactions more explicitly.

We agree with the reviewer that the surface charge representation of Rip1 is somewhat hard to interpret. We thus followed the reviewers' suggestion and show now the side chains for the negatively charged residues in Rip1 that form contacts to Bcs1 in the revised Figure 2C.

2. While the mutagenesis experiments on Bcs1 establish the importance of these residues, it would be valuable to determine whether mutations in Rip1 could produce similar lethal phenotypes.

We appreciate this suggestion from the reviewer and, accordingly, have performed Rip1 mutations. We find that single charge inversion mutations in Rip1 (D139R,

D145R, E201R and D210R) don't show any growth defect while double mutations indeed show lethal phenotypes on non-fermentable carbon source. We added these new findings to the revised manuscript and to revised Figure 2D. We further added a new Appendix Figure S6 summarizing our complete mutational analysis of Bcs1 and Rip1.

3. The authors should clarify why the TM helices adopt either defined or less ordered conformations across different states. Is there a coordinated relationship between TM helix movement and other domains of Bcs1?

The manuscript has indeed been somewhat vague on this point in the manuscript. Here, we only compare the two ATP- γ S states, where we observe that the TD helices are well defined in the ATP- γ S1 state but less defined in ATP- γ S2. Upon comparing these two states we describe a rotation of the Bcs1 middle domain versus the AAA domain (Figure 3D, 3E) and state that the TD in ATP- γ S2 appears to follow the rotation of the middle domains as a rigid body based on the position of the observable short parts of the helices. While we cannot fully explain how this rotation leads to more flexibility of the TD, we added a sentence to the revised manuscript stating that this rotation of the TM helices apparently corresponds to destabilization of the TD.

To assess a general correlation between TM helix movement and Bcs1 state, we carefully evaluated all the currently available cryo-EM structures (see table below; also as Appendix Table S2) and classified the TD helices as “resolved”, “partly resolved” or “not resolved”. Here, we noted that visibility (conformational stability) of the TM helices is rather poorly correlated with any state of Bcs1. We rather think that this highly depends on the environment (property of the lipid/detergent micelle) and quality of the reconstructions in the TD area, and this is mainly driven by technical parameters like signal-to-noise ratio, the number of particles, ice thickness, thoroughness of 3D classification, etc. For example, some mouse Bcs1 reconstructions show only density of the TD helices after re-processing/increasing particle amount. While the first published mBCS1-Apo state didn't show a defined TD, a later reconstruction showed a more defined TD. We made similar observations with our yeast datasets. Moreover, all structures are derived from Bcs1 samples in detergent, the use of which may further influence instability of TD helices in the mixed detergent/lipid micelles.

Given this complexity, we decided to only compare the TD densities for well-resolved classes that were derived from the same dataset, namely ATP- γ S1 and ATP- γ S2. Here, we also find our observations in agreement with the well-resolved mBCS1-ATP states, for which a similar delocalization of TD helices was described.

Species	State	PDB	EMDB	TM-helices	Symmetry	Reference
S. cerevisiae	Bcs1-ADP	6SH3	10192	Resolved	C7	(Kater et al. , 2020)
S. cerevisiae	Bcs1-Apo1	6SH4	10193	Resolved	C7	(Kater et al. , 2020)
S. cerevisiae	Bcs1-Apo2	6SH5	10194	Partially resolved	C7	(Kater et al. , 2020)
S.	Bcs1-	-	51561	Partially	C1	This study

cerevisiae	Apo1- Rip1-FeS			resolved		
S. cerevisiae	Bcs1- Apo1- Rip1-TM	9GS2	51537	Partially resolved	C1	This study
S. cerevisiae	Bcs1- ATPgS1	-	53185	Resolved	C1	This study
S. cerevisiae	Bcs1- ATPgS1	9GSN	51552	Resolved	C7	This study
S. cerevisiae	Bcs1- ATPgS2	-	53189	Partially resolved	C1	This study
S. cerevisiae	Bcs1- ATPgS2	9GU9	51605	Partially resolved	C7	This study
S. cerevisiae	Bcs1- ATPgS2- Rip1-FeS	-	51562	Not resolved	C1	This study
M. musculus	Bcs1-Apo	6UKP	20808	Not resolved	C7	(Tang et al. , 2020)
M. musculus	Bcs1- ATPgS	6UKS	20811	Not resolved	C7	(Tang et al. , 2020)
M. musculus	Bcs1- ATP1	8TI0	41276	Not resolved	C1	(Zhan et al. , 2024)
M. musculus	Bcs1- ATP1	8T5U	41061	Not resolved	C7	(Zhan et al. , 2024)
M. musculus	Bcs1- ATP2	8TPL	41476	Partially resolved	C1	(Zhan et al. , 2024)
M. musculus	Bcs1- ATP2	8TP1	41462	Resolved	C7	(Zhan et al. , 2024)
M. musculus	Bcs1- ADP	8T7U	41095	Resolved	C1	(Zhan et al. , 2024)
M. musculus	Bcs1- ADP	8T14	40954	Resolved	C7	(Zhan et al. , 2024)
M. musculus	Bcs1- Rip1	-	41609	Not resolved	C1	(Zhan et al. , 2024)
M. musculus	Bcs1-Apo	8TBY	41148	Partially resolved	C1	(Zhan et al. , 2024)

It is likely that binding of the Rieske substrate leads to a delocalization of the TD helices, since the helices were invisible in both yeast and mouse reconstructions of substrate-bound Apo Bcs1, but we would still refrain from mentioning this observed tendency in the manuscript because these substrate-bound reconstructions display lower resolution and contain lower amounts of particles compared to the substrate-lacking structures.

4. The presentation of Fig. 4C is potentially misleading, as it suggests that the TM helices are ordered in this state. However, this appears inconsistent with the text and with Fig. 4B, which shows a lack of ordered TM density.

We agree with the reviewer on this point. However, we have not modified Fig.4C showing the electrostatic charge distribution but clarified in the legend that these regions are not resolved and just show for illustrative purposes the position of the TD as in the ATP γ S1 state. In addition, we added a new panel (Figure 4D) to make it clearer that the TD (now shown in transparent) in the substrate-bound Bcs1- ATP- γ S2 state is delocalized.

5. The authors conclude that the translocated Rip1 within the inner membrane vestibule is less ordered, precluding model fitting. However, the density appears to have a characteristic crescent shape (side views in Fig. 4B & C). Would this shape preclude the ability to rigid body fit a Rip1 model into the density?

When analyzing our reconstructions we initially had the same impression and hoped to be able to dock the substrate. However, the Rip1 density in our translocating state is indeed not very well resolved and most likely represents an average of multiple positions of the substrate inside the IM vestibule. This may lead to the observed crescent shape that is somewhat different to the well resolved more oval shape of Rip1 in the binding state. Despite the lack of resolution, we already attempted to fit Rip1, and what is shown in Figure 4C of the original manuscript is a surface representation of one possible model colored according to charge.

However, even though the dimensions and the shape are consistent with the Rieske protein, we cannot unambiguously orient Rip1. We show in Fig EV5 two of several possible orientations, one with the Rip1 N-terminus facing the IMS side, one facing towards the matrix. We favor the latter, because it positions the negatively charged surface of Rip1 towards the TD basket (IM vestibule), that is lined by complementary positive charges. The validity of this overall positioning is supported by our mutational analysis of residues R69 and R81, located inside the IM vestibule (see also response to reviewer 2). Here especially R69 showed a lethal phenotype on non-fermentable carbon source indicating its importance for Rip1 translocation and complex III assembly (shown in Fig. 4E).

In conclusion, while unambiguous docking is unfortunately not possible, we have a most plausible scenario which we display now in the Fig EV5.

6. Given the poor local resolution of translocated Rip1, further image processing might enhance the structural details. If particle alignment is driven by the matrix side of the complex, would focused refinement around the middle domains improve local resolution in this region?

We agree with the reviewer that this strategy can often lead to improved density features. We tried focused classifications and refinements as well as 3D variability analysis in CryoSPARC and RELION using masks covering the Rip1-bound IM vestibule. Unfortunately, none of these tools led to a significant improvement in the resolution of the substrate density. We conclude that Rip1 in the IM vestibule explores a continuous conformational space resulting in heterogeneity that is very difficult to classify further (also due to the limited signal for the small protein). From a biological point of view, such heterogeneity would be expected because the substrate is already translocated and ready to be released. Therefore, distinct interactions between Rip1 and Bcs1, as required initially for substrate recognition and loading into the matrix vestibule, are not necessary and likely unspecific at this stage.

7. It would be interesting to examine whether the Bcs1-Rip1 interactions are conserved across species. The authors provide an alignment of Bcs1, highlighting the EWR and DDR motifs (Fig. EV10), but do the interacting residues in Rip1 exhibit

similar conservation? Investigating this could provide co-evolutionary insights into why Rip1 remains the sole substrate of Bcs1.

We followed the reviewers' suggestion and examined the conservation of Bcs1-interacting residues in Rip1. A multiple sequence alignment of Rip1 homologs from species that use both Bcs1 (in mitochondria) and the Tat pathway (petC in chloroplasts or qcrA in prokaryotes) was performed. The alignment shows that the Bcs1-interacting residues D139, D145 and E201 are conserved only in cases where Bcs1 is required for translocation. In contrast only D139 is conserved in chloroplasts and none of the three residues are conserved in prokaryotes. Moreover, we noted that these residues are exposed to the surface when Rip1 is integrated into complex III, indicating that they are not under evolutionary pressure for complex III interaction or activity. We added a short paragraph summarizing these results in the revised manuscript and added a new Fig EV4.

Other Comments:

1. The authors should clarify whether the ~30 Å diameter opening in the middle domains in ATP_S-bound states is sufficient to accommodate Rip1 translocation.

As suggested by the reviewer, we now explicitly state the dimensions of Rip1 (about 30 Å) in the main text and we added bar indicating Rip1's size in Figure 4B. As shown in Figure 3B, the overall change in conformation of Bcs1 leads to an opening of the middle domain from 15 Å to 30 Å, only the latter being sufficient to accommodate Rip1 for translocation.

2. There are several typographical errors throughout the manuscript, including:

"Bsc1" (page 9, heading)

"dfference" (page 12)

"postion" (page 14)

"disitinct" (page 15)

"concommittent" (page 15)

We thank the reviewer for pointing this out and have corrected it in the revised manuscript.

Overall, this is a well-executed study that significantly advances our understanding of Bcs1 function. I look forward to seeing the authors' revisions.

We are pleased to read that the reviewer has found our work significant and are thankful for this constructive assessment of our work.

Referee #2:

This manuscript by the Beckmann group addresses an exceptionally fascinating conceptual problem in cellular biochemistry, the translocation of folded macromolecules across biological membranes. Studies on bacterial and peroxisomal systems as well as the nuclear pore have revealed that intrinsically disordered protein domains within translocator complexes represent an important feature of the

passageway for folded proteins across membranes. Yet another, highly conserved and crucial factor of the mitochondrial inner membrane, the Rieske-Fe/S protein (Rip1), which requires co-factor binding prior to the final export step from the mitochondrial matrix, was shown to depend on a non-canonical AAA ATPase, termed Bcs1, for translocation. Despite many studies on the molecular mechanisms of Rip1 biogenesis, the mechanism of Bcs1-mediated Rip1 translocation has remained enigmatic.

Beckmann and colleagues show here for the first time cryo-EM images of the Rip1-loaded homo-heptameric Bcs1 complex at distinct stages of translocation in an excellent resolution that allows the visualization of molecular details and translocase-substrate interactions. The comparison of a substrate-docked pre-translocation state in the absence of nucleotides is thoroughly compared to advanced translocation intermediates formed in the presence of non-hydrolysable ATP analogues. In this way, the transition from substrate binding via the matrix-localized ATPase domain to occlusion in a Bcs1-specific middle domain and further to a localization within the membrane-embedded domain can be beautifully recapitulated.

The quality of the data is impressive and the manuscript is very well written, peppered with sophisticated details, but also accessible for non-expert readers. It certainly represents a major conceptual advance in an important area of cellular biochemistry and is well suited for publication in the EMBO Journal.

We thank the reviewer for this positive view on our work.

However, the following two major points should be convincingly addressed by the authors prior to acceptance:

1. In mitochondria, insertion of the Fe-S cluster occurs before Bcs1-mediated translocation of Rip1 and co-factor bound Rip1 is the clearly preferred substrate for Bcs1, which is also acknowledged by the authors in the discussion. They even speculate, how co-factor incorporation may optimize substrate interaction with the ATPase domain. Given the advanced possibilities of molecular modeling, the authors are strongly encouraged to try and model a Fe-S cluster-bound holo-state of Rip1 into their structures of Rip1-Bcs1 binding and translocation complexes to test their hypothesis.

First, we must apologize for not being clear enough in our original manuscript leading to a potential misunderstanding. We already modeled the 2Fe-2S cluster (shown in Fig. 1D and 2B) and showed a fit into our density of Bcs1-Apo-Rip1-TM (Appendix Fig S4). The reviewer may have been misled by our ball representation of Fe and S atoms (see also response to the first “minor point”). We now explicitly describe this in the figure legends. In addition, we added a new Figure EV3 to show how the cluster nicely fits the characteristic density in the Bcs1-Apo-Rip1-TM map at higher contour levels (typical feature of electron-dense/high atomic number Fe-S clusters) and compared it to the cluster-bound Rip1 from active CIII complexes studied by cryo-EM (PDBs 6HU9 and 6GIQ).

Unfortunately, we cannot properly dock Rip1 or resolve the cluster of Rip1 in the translocation complex.

2. When discussing the driving forces for Rip1 to move on from early to later translocation stages, besides conformational changes in Bcs1, the authors suggest a role for positively charged amino acid residues (R69, R81) in the IM vestibule that may attract Rip1. The initial interaction between Rip1 and Bcs1 on the matrix site of the membrane in the pre-translocation complex is also suggested to be mediated by electrostatic interactions. Does the proposed transition make sense from the energetic perspective? The authors should analyze Bcs1 variants R69 and R81 with suitable amino acid substitutions. As altered substrate interactions and/or conformational state appear to affect the migration behavior of Rip1-Bcs1 complexes in native gels, this may offer a possibility to examine in vitro, if R69 and R81 are important for such transitions. At least, the authors should provide in vivo growth test for the respective bcs1 mutants as shown in Figure 2 to substantiate the role of the proposed docking site amino acid cluster

We followed the suggestion of the reviewer and analyzed mutants of the two arginines in our growth test. We observe that R69A and R69E mutations lead to a lethal phenotype on non-fermentable carbon source, and R81 mutations show a detectable growth defect. We therefore conclude that the mechanism we suggest does indeed make sense, and that besides the conformational changes of the entire Bcs1 complex upon ATP binding also the positive charges in the IM vestibule contribute to the translocation of Rip1 from the matrix into the IM vestibule.

We added these new results to a new panel in revised Figure 4 (Figure 4E). We also added a new Appendix Figure S6 summarizes the results for all the mutagenesis experiments done on Bcs1 and Rip1.

Minor points:

- I may have overlooked this, but what does the yellow ball within the Rip1 FeS domain that is visible in many structural models actually represent?

We apologize for this misleading representation. The yellow and the dark red balls represent Sulphur (S) and iron atoms (Fe) of the 2Fe-2S cluster domain. We now explicitly describe this in the figure legends (see also our response to major point 1).

- Out of curiosity: A study by the Trumpower group has identified a structurally important disulfide bond in folded Rip1 between C164 and C180. Is this disulfide present in the structures of Rip1 translocation intermediates shown here?

Although we are not able to resolve bonds in our map, we find density in the region of this disulfide bond as shown in the new Figure EV3. This density is in full agreement with the shorter distance of the two cysteine residues (and the peptide backbone) in the presence of a disulfide bond. In addition, we observed that using β -mercaptoethanol in non-denaturing concentrations (1 mM) in our purification results in Rip1 protein that cannot bind to Bcs1 anymore, as shown in the blue native gel in panel Fig. EV3B. Consequently, we note that the presence of the disulfide bond is necessary for Bcs1 substrate recognition and should be present in our structures. This is mentioned in the revised text.

- In the introduction the authors state that Rip1 incorporation into complex III is a prerequisite for dimerization and super-complex formation and cite a paper from the year 2000. This is not anymore the current state of knowledge. Several groups have shown dimer formation of complex III and even super-complex formation (although inefficient) of intermediates lacking Rip1.

We thank the reviewer for the hint that this paper is no longer current state of knowledge. We have expanded the list of references and commented on the ability of complex III to oligomerize into super-complex intermediates in absence of Rip1 in the introduction.

Referee #3:

Bcs1, an AAA-ATPase in the mitochondrial inner membrane (IM), is a unique translocator that mediates the transfer of a ligand-bound folded protein, Rieske FeS protein (Rip1 in yeast), from the matrix to the intermembrane space (IMS) across the IM without its unfolding. Although high-resolution cryo-EM structures of substrate-free Bcs1 and a low-resolution structure of Bcs1 bound to a substrate (Zhan et al., 2024) are available, the precise mechanism by which folded Rip1 is translocated through Bcs1 across the IM still remains elusive. In this manuscript, the authors purified full-length Bcs1 from yeast and Rip1 derivatives (Rip1-FeS and Rip1-TM) from *E. coli* and optimized conditions to reconstitute yeast Bcs1-Rip1 derivative complexes. They determined high-resolution cryo-EM structures of Bcs1(Apo)-Rip1, Bcs1(ATP γ S), and Bcs1(ATP γ S)-Rip1, and on the basis of these structures, they proposed a model for substrate transfer from the matrix toward IMS via Bcs1. In this model, Rip1 binds to nucleotide-free Bcs1 (loading step). Then, ATP binding to Bcs1 induces a conformational change, a constricting the matrix vestibule and simultaneously opening the seal pore between the matrix and IM vestibules, leading to substrate translocation to the IM vestibule (gating step). The reported structures, especially those of Bcs1 bound to a substrate are novel, and combined with the analysis of Bcs1-Rip1 complex formation, these findings significantly advance our understanding of how Bcs1 translocates a folded substrate across the IM without ion leakage, making this study of broad interest to NSMB readers.

We thank the reviewer for the positive comments.

However, I have the following concerns for the authors to consider.

- Based on in vitro reconstitution under different nucleotide conditions, the authors suggest that Rip1 binds to nucleotide-free Bcs1. However, given the high cellular ATP concentrations, I am skeptical about the presence of a significant fraction of nucleotide-free Bcs1 waiting for a substrate. While Fig. 4A demonstrates efficient Bcs1-Rip1 complex formation, this does not necessarily mean that ATP-bound Bcs1 cannot accommodate Rip1 under conditions (e.g., varying ATP concentrations).

We agree with the reviewer, and we are wondering ourselves – given the high cellular ATP concentrations – if Bcs1 exists for a longer period in the apo state in vivo. However, we never observed Rip1 engaging Bcs1 in any conditions that have a nucleotide bound; we and others found that ATP-loaded Bcs1 adopts a conformation

that precludes Rip1 binding from the matrix side, the opening is simply too small (see Fig 3B). We also tried to load Bcs1-ADP with Rip1 and performed cryo-EM analysis of this sample in the presence of excess ADP, but we never observed density for Rip1. We thus conclude that such an apo state must also exist in vivo, even though it may be very short-lived. One possibility is that after one round of Rip1 translocation and ATP hydrolysis Bcs1 is left with ADP which dissociated upon Rip1 binding for the next round of translocation.

- Zhan et al (2024) showed that both the ADP-bound and nucleotide-free forms of mouse Bcs1 can form a complex with Rieske FeS protein. Does this discrepancy simply reflect a species difference?

Yeast and mouse Bcs1 indeed appear to be different with respect to the Apo and ADP states. While in mBCS1-Apo and ADP adopt a very similar conformation, the yBcs1-ADP state clearly differs from the Apo state (see Fig EV1) and, in addition, yeast Bcs1 can adopt two different Apo states, Apo1 and Apo2. For this reason, it is not clear, whether Rieske binds the ADP or Apo form of mBCS1 because at the given resolution the authors could not unambiguously distinguish ADP from Apo states. A clear differentiation between the two can only be achieved by checking the nucleotide binding pockets with a map at sufficiently high resolution.

In any case, it appears plausible that mammalian BCS1 and yeast Bcs1 complexes employ slightly different translocation mechanisms, since in contrast to the yeast Rip1 the mammalian Rieske contains in addition to the transmembrane helix a globular matrix domain. This domain, which is connected to the conserved Rieske domain via the TM helix, requires for release a complete lateral opening between subunits of the mammalian BCS1 complex, whereas the yeast Bcs1 complex can remain fully assembled while opening of the IM basket TM helices would be sufficient for releasing the Rip1 TM domain into the IMM.

- In Fig. 1A, is the lower band on the BN gel a Bcs1 monomer?

We apologise for not labeling this band properly. It represents the Rip1 substrate. We added label now for revised Fig. 1A.

- In Fig. 1C, how reliable is the TD part in the Bcs1(Apo)heptamer-Rip1 complex? Does this form indeed lack nucleotides? It is important to show the EM density map around the nucleotide binding sites to confirm the absence of residual ADP retained during purification. Alternatively, residual nucleotides, if present, can be detected by UV measurements after heat denaturation of the Bcs1 sample used for EM measurements.

The TD part in our CryoEM map for Bcs1(Apo)heptamer-Rip1 complex is only partially resolved and we only showed that model of the TD based on the Bcs1 Apo1. We made it now clearer in the legend that the helices are shown in transparency to represent this flexibility. We added a more general discussion of visibility of the TD helices in cryo-EM maps of Bcs1 in Appendix Table S2 (see also response to reviewer 1, point 3).

Further, we followed the reviewers' suggestion and show the EM density map around the nucleotide binding sites in a new Figure EV2. When comparing the densities around the ATP binding pocket of each protomer in the Bcs1-Apo-FeS state (this study) to that of the protomers in the Bcs1-ADP state (PDB: 6SH3; EMDB: 10192) we clearly observe that ADP density is absent in all seven binding pockets of Bcs1 (Figure E2VA-B).

- In Fig. 2, the authors should describe the interactions between Bcs1 and Rip1 through conserved motifs and assess their importance by mutagenesis. Why were conserved residues W213 and D300 not discussed?

As suggested by the reviewer, we expanded our mutational analysis with residues D300 and D301 (DDR motif) (see revised Figure 2D) and W213 (EWR motif; see Appendix Figure S6) and incorporated suggestions from Reviewer #1. Mutation of all three residues lead to a lethal phenotype on non-fermentable carbon source confirming their importance. We didn't discuss W213 in the original manuscript because our structure showed that this residue does not contribute to the interaction surface with Rip1. We think that it most likely plays a role in maintaining the structural integrity of the EWR motif, explaining its lethal phenotype.

In the central panel in Fig. 2C, should R212 be R214?

We thank the reviewer for noticing the typographical mistake. We changed revised Figure 2C accordingly.

- In Figs. 3 and 4, ATP γ S density should be shown to confirm the presence of the Bcs1(ATP γ S) form.

We now show density for ATP γ S to confirm the presence of nucleotide in Bcs1 for all 3 ATP γ S states in new Figure EV2, panel C-E.

- In Fig. EV1, how was the binding of Fe-S to the purified Rip1 verified?

Since we clearly identified the cluster in our density map of the Bcs-Apo1-Rip1-TM sample, we did not further verify experimentally whether the 2Fe-2S cluster is present in our purified Rip1. Moreover, the observed density for the cluster in our structure resembles the one observed for Rip1 incorporated into the active complex III (PDB:6HU9, EMDB: 0262; as well as PDB:6GIQ, EMDB:0004). We have added a new Figure EV3 to display this in more detail.

- Page 7, line 26: 79.156 is 106,497?

We thank the reviewer for noticing the typographical mistake. It has been corrected.

- Page 7, line 27: 57.257 is 57,257?

We thank the reviewer for noticing the typographical mistake. It has been corrected.

- Page 9, line 17: Bsc1 should read Bcs1.

We thank the reviewer for noticing the typographical mistake. It has been corrected.

- Page 12 (for Fig. 4A): the BN-gel lanes should be labeled clearly and referred to by lane numbers in the text for better readability.

As suggested by the reviewer, we have updated the figure and the legend, and the revised manuscript improve readability, accordingly.

- Page 16, line 10: analogon is analog?

We thank the reviewer for noticing the typographical mistake. It has been corrected.

- Page 22, line 2: Increase 10/30 GL is Increase 10/300 GL?

We thank the reviewer for noticing the typographical mistake. It has been corrected.

Prof. Roland Beckmann
Gene Center Munich, Department of Biochemistry, University of Munich, Germany
Biochemistry
Feodor-Lynen-Str. 25
Munich, Bavaria 81377
Germany

22nd Apr 2025

Re: EMBOJ-2024-119918R
Mechanistic insights into Bcs1-mediated translocation of the folded Rieske protein

Dear Roland,

Thank you for submitting your revised manuscript to The EMBO Journal. Two of the original referees have now assessed it once more, and I am happy to say that both were fully satisfied with the revisions and have no further concerns at this stage. After incorporation of the following remaining editorial issues, we should therefore be able to proceed with formal acceptance of the study:

- Please correct the Section order as follows: Title page - Abstract & Keywords - Introduction - Results - Discussion - Methods - Data Availability - Acknowledgements - Disclosure and Competing Interests Statement - References - Figure Legends - Table(s) - Expanded View Figure Legends.

- In order to make the title slightly more explicit and accessible to a broad readership, I would propose to still incorporate a few keywords, to read e.g.

Mechanistic insights into Bcs1-mediated MITOCHONDRIAL translocation of the folded Rieske protein
Mechanistic insights into Bcs1-mediated MITOCHONDRIAL MEMBRANE translocation of the folded Rieske protein
STRUCTURAL insights into Bcs1-mediated MITOCHONDRIAL MEMBRANE translocation of the folded Rieske protein

- Since the yeast growth assays displayed in Figures 2 and 4 all appear to be excerpts of the comprehensive combined display in Appendix Figure S6, please make sure to explicitly indicate this (and the underlying rationale) in all three respective figure legends.

- As we are switching from a free-text author contribution statement towards a more formal statement based on Contributor Role Taxonomy (CRediT) terms, please remove the present Author Contribution section and instead specify each author's contribution(s) directly in the Author Information page of our submission system during upload of the final manuscript. See <https://casrai.org/credit/> for more information.

- Finally, please provide suggestions for a short 'blurb' text prefacing and summing up the study in two sentences (max. 250 characters), followed by 3-5 one-sentence 'bullet points' with brief factual statements of key results of the paper; they will form the basis of an editor-written 'Synopsis' accompanying the online version of the article. Please also upload a synopsis image, which can be used as a "visual title" for the synopsis section of your paper. The image should be in PNG or JPG format with the modest dimensions of 550 x 300-600 pixels (width x height). Maybe a re-arranged version of Figure 5 (with steps 1-2-3 arranged vertically on top of each other?) could be used?

I am returning the manuscript to you for a final round of revision, solely to allow you to make these modifications and upload the revised files. Once we will have received them, we should be ready to swiftly proceed with formal acceptance and production of the manuscript.

Thank you for the opportunity to consider your work for publication. I look forward to your revision.

With kind regards,

Hartmut

9) To facilitate reproducibility and cross-laboratory adoption of methodologies, please structure the Materials & Methods section as outlined in our guide to authors, including a completed Reagents and Tools Table that can be downloaded from our author guidelines as well (<https://www.embopress.org/page/journal/14602075/authorguide#structuredmethods>).

10) Digital image enhancement is acceptable practice, as long as it accurately represents the original data and conforms to community standards. If a figure has been subjected to significant electronic manipulation, this must be clearly noted in the figure legend and/or the 'Materials and Methods' section. The editors reserve the right to request original versions of figures and the original images that were used to assemble the figure. Finally, we generally encourage uploading of numerical as well as gel/blot image source data; for details see: embopress.org/page/journal/14602075/authorguide#sourcedata

Further information is available in our Guide For Authors:

In the interest of ensuring the conceptual advance provided by the work, we recommend submitting a revision within 3 months (21st Jul 2025). Please discuss the revision progress ahead of this time with the editor if you require more time to complete the revisions. Use the link below to submit your revision:

Link Not Available

Referee #1:

The revised manuscript has addressed my concerns and I find it suitable for publication. The work is excellent.

Referee #2:

The authors have adequately addressed all points raised by the reviewers. In my view, the revised manuscript can be accepted for publication in the EMBO Journal.